# Adult medial habenula neurons require GDNF receptor GFRα1 for synaptic stability and function

Diana Fernández-Suárez[1,2]*, Favio A. Krapacher[1], Katarzyna Pietrajtis[3], Annika Andersson[1], Lilian Kisiswa[1,4], Alvaro Carrier-Ruiz[1], Marco A. Diana[3], Carlos F. Ibáñez[1,2,5]*

1 Department of Neuroscience, Karolinska Institute, Stockholm, Sweden, 2 Department of Physiology and Life Sciences Institute, National University of Singapore, Singapore, Singapore, 3 Sorbonne Université, CNRS, INSERM, Neurosciences Paris Seine–Institut de Biologie Paris Seine (NPS-IBPS), Paris, France, 4 Department of Biomedicine, Aarhus University, Aarhus C, Denmark, 5 Peking-Tsinghua Center for Life Sciences, PKU-IDG/McGovern Institute for Brain Research, Peking University School of Life Sciences and Chinese Institute for Brain Research, Beijing, China

* diana.fernandez.suarez@ki.se (DF-S); carlos.ibanez@ki.se (CFI)

**Data Availability Statement:** All the data files used to create the figures of this manuscript are available from the Figshare database (accesion number 2021/123406) and can be found in the

## Abstract

The medial habenula (mHb) is an understudied small brain nucleus linking forebrain and midbrain structures controlling anxiety and fear behaviors. The mechanisms that maintain the structural and functional integrity of mHb neurons and their synapses remain unknown. Using spatiotemporally controlled Cre-mediated recombination in adult mice, we found that the glial cell–derived neurotrophic factor receptor alpha 1 (GFRα1) is required in adult mHb neurons for synaptic stability and function. mHb neurons express some of the highest levels of GFRα1 in the mouse brain, and acute ablation of GFRα1 results in loss of septohabenular and habenulointerpeduncular glutamatergic synapses, with the remaining synapses displaying reduced numbers of presynaptic vesicles. Chemo- and optogenetic studies in mice lacking GFRα1 revealed impaired circuit connectivity, reduced AMPA receptor postsynaptic currents, and abnormally low rectification index (R.I.) of AMPARs, suggesting reduced $Ca^{2+}$ permeability. Further biochemical and proximity ligation assay (PLA) studies defined the presence of GluA1/GluA2 ($Ca^{2+}$ impermeable) as well as GluA1/GluA4 ($Ca^{2+}$ permeable) AMPAR complexes in mHb neurons, as well as clear differences in the levels and association of AMPAR subunits with mHb neurons lacking GFRα1. Finally, acute loss of GFRα1 in adult mHb neurons reduced anxiety-like behavior and potentiated context-based fear responses, phenocopying the effects of lesions to septal projections to the mHb. These results uncover an unexpected function for GFRα1 in the maintenance and function of adult glutamatergic synapses and reveal a potential new mechanism for regulating synaptic plasticity in the septohabenulointerpeduncular pathway and attuning of anxiety and fear behaviors.

following link: https://figshare.com/projects/Raw_Data_Fernandez-Suarez_et_al_2021/123406.

**Funding:** Financial support for this research was provided by grants to C.F.I. from the Swedish Research Council (https://www.vr.se; 2016-01538 and 2020-01923) and the National Research Foundation of Singapore (https://www.nrf.gov.sg; R-711-000-052-281) and to D.F.S. from Karolinska Institutet (www.ki.se; 2016fobi50068). The funders had no role in study design, data collection and analysis, decision to publish or preparation of the manuscript.

**Competing interests:** The authors have declared that no competing interests exist.

**Abbreviations:** AP, anteroposterior; BAC, bed nucleus of the anterior commissure; ChAT, choline acetyltransferase; ChR2, Channelrhodopsin-2; CNO, clozapine N-oxide; CNS, central nervous system; CR, calretinin; DREADD, designer receptor exclusively activated by designer drug; dmHb, dorsal medial habenula; DV, dorsoventral; EPSC, excitatory postsynaptic current; FR, fasciculus retroflexus; GDNF, glial cell line–derived neurotrophic factor; GFRα1, glial cell–derived neurotrophic factor receptor alpha 1; Het, heterozygous; HRP, horseradish peroxidase; i.p, intraperitoneally; IPN, interpeduncular nucleus; KO, knockout; LHb, lateral habenula; mHb, medial habenula; ML, mediolateral; NCAM, neural cell adhesion molecule; NDS, normal donkey serum; PFA, paraformaldehyde; PLA, proximity ligation assay; PSD, postsynaptic density; PVDF, polyvinylidene fluoride; R.I, rectification index; RT, room temperature; RT-PCR, reverse transcription PCR; SNpc, substantia nigra pars compacta; SP, substance P; TS, triangular septum; VGAT, vesicular GABA transporter; VGlut1, vesicular glutamate transporter 1; VGlut2, vesicular glutamate transporter 2; vmHb, ventral medial habenula; VTA, ventral tegmental area; WT, wild-type.

## Introduction

The habenula is a phylogenetically conserved brain structure, present in all vertebrates, consisting of 2 small nuclei located above the thalamus, close to the midline [1]. Due to its distinct location and connectivity, it is thought to function as a node linking more recently evolved forebrain structures involved in executive functions with ancient midbrain areas that process aversion and reward [1]. Lesions to the habenula affect different behaviors related to stress, fear, aversion, anxiety, pain, and sleep [2–5]. In mammals, the habenula is subdivided into 2 distinct subnuclei, the lateral habenula (LHb) and medial (mHb) habenula, which present different neurochemical properties, connectivity, and functions [1]. In the fasciculus retroflexus (FR), one of the longest major fiber tracts in the brain and among the first to form during development, afferent axons from the 2 subnuclei are segregated, with LHb axons in the periphery, innervating different midbrain structures, and mHb axons in the center, all terminating in the interpeduncular nucleus (IPN) [1]. Perhaps due to its prominent connections to dopaminergic and serotoninergic neurons in ventral tegmental area (VTA), substantia nigra pars compacta (SNpc), and raphe nuclei, the LHb has been more extensively studied than the mHb [6]. However, mHb neurons display several characteristics that make them quite unique in the mammalian brain. A striking property of mHb neurons is their spontaneous pacemaking activity, which generates tonic trains of action potentials of about 2 to 10 Hz [7]. Moreover, GABAergic inputs from the medial septum elicit excitation, rather than inhibition, in mHb neurons due to lack of the KCl co-transporter KCC2, a feature that is more common in the developing brain, but rare among adult central nervous system (CNS) neurons [8]. Similarly, GABA released by postsynaptic IPN neurons has been shown to act as a retrograde messenger on presynaptic GABA-B receptors in mHb terminals to amplify glutamatergic transmission at habenulointerpeduncular synapses [5,9]. Furthermore, the mHb is the only known structure in the adult brain that contains functional NMDA receptor GluN1/GluN3A, through which glycine, otherwise known as a major inhibitory neurotransmitter, regulates neuronal excitability in mHb neurons [10]. Finally, while the majority of AMPA receptors in the adult brain are $Ca^{2+}$ impermeable [11], inputs to the mHb from the triangular septum (TS) and bed nucleus of the anterior commissure (BAC), as well as the mHb output to the IPN, produce glutamatergic responses through $Ca^{2+}$-permeable AMPA receptors [9,12]. Despite its unique properties and functional importance in fear and anxiety, very little is known about the mechanisms that regulate the structural and functional plasticity of mHb neurons and their synapses [6]. Moreover, aside from electrophysiological evidence [9,12], the molecular composition and functional importance of mHb $Ca^{2+}$-permeable AMPA receptors remain to be established.

Glial cell–derived neurotrophic factor receptor alpha 1 (GFRα1) is the main binding receptor for glial cell line–derived neurotrophic factor (GDNF) [13]. It lacks an intracellular domain and is anchored to the plasma membrane through a glycosyl phosphatidylinositol link [14]. Through the action of plasma membrane lipases, glycosyl phosphatidy cleavage results in receptor shedding from the cell surface, allowing GFRα1 to act on nearby cells [15,16]. GFRα1 has been shown to function both in conjunction with transmembrane co-receptor subunits, such as the receptor tyrosine kinase RET [14] and the neural cell adhesion molecule (NCAM) [17], as well as independently of these receptors [18]. During development, GDNF and GFRα1 are essential molecular determinants for neuronal survival, migration, and differentiation of a variety of neuronal subpopulations in the peripheral and central nervous systems [19]. Exogenously provided GDNF can promote survival of midbrain dopaminergic neurons, both in vitro and in animal models of Parkinson disease and therefore has been tested in clinical trials for that disease [20]. On the other hand, the role of endogenous GDNF for the survival of adult dopaminergic neurons is less clear, as both evidence for and against has been provided

[21–24]. Although GFRα1 is highly expressed in several areas of the adult brain, such as the septum, midbrain, and habenula [25–28], its function in the adult nervous system remains unknown.

The prominent expression of GFRα1 in the adult mHb led us to investigate its function in these neurons using temporally and spatially controlled Cre-mediated inactivation of the *Gfra1* gene. In this study, we report a previously unknown and essential role for GFRα1 in the maintenance, integrity, and function of mHb synapses in the adult septum→mHb→IPN pathway. Significantly, loss of GFRα1 in the mHb resulted in changes in the molecular composition of mHb AMPA receptors that reduced their $Ca^{2+}$ permeability, as well as alterations in fear and anxiety behaviors that paralleled those observed after lesion of the septohabenular pathway [4].

## Results

### Characterization of GFRα1 and co-receptor expression in the septum→Hb→IPN pathway

Inspection of sections through the brain of tamoxifen-injected 3-month-old mice carrying a *Rosa26*[dTOM] reporter under the control of a *Gfra1*[Cre-ERT2] allele (herein referred to as *Gfra1*[d-TOM]) revealed very high dTOM signal in the mHb and the FR (Fig 1A), comparable to or exceeding that observed in better known sites of GFRα1 expression, such as septum, SNpc, or VTA (S1A Fig). Particularly striking was the intensity of dTOM signal in mHb axons of the FR and their terminals in the IPN (Fig 1A, S1A Fig), in agreement with previous studies [25,26,28]. Real-time reverse transcription PCR (RT-PCR) analysis of *Gfra1* mRNA expression in regions microdissected from adult mouse brain showed highest expression in the Hb (S1B Fig), while western blots confirmed highest expression in the IPN (S1D Fig), most likely derived from mHb axon terminals, since *Gfra1* mRNA was at background levels in this structure (S1B Fig). Expression of *Gdnf* mRNA (S1C Fig) and GDNF protein (S1E Fig) was found across a number of adult brain regions, including Hb, septum, and IPN. As GDNF is a soluble, diffusible ligand, it is likely to be available to most neurons in those structures.

For a closer inspection of the cellular distribution of the expression of GFRα1 and its co-receptors in the adult Hb, IPN, and septum, we used mouse strains expressing GFP from either the *Gfra1* (*Gfra1*[GFP]) (see Methods section) or *RET* (*RET*[GFP]) [29] loci, in addition to *Gfra1*[d-TOM] mice and immunohistochemistry and RNAscope for endogenous GFRα1 and NCAM. All neurons in the mHb were positive for *Gfra1*[GFP], while no GFP signal could be detected in the LHb (Fig 1B and 1C) in agreement with previous studies [26]. *Gfra1*[GFP] signal co-localized with choline acetyltransferase (ChAT) in the ventral medial habenula (vmHb) (Fig 1B and 1C). As it is well known, substance P (SP) can be detected at very low levels in dorsal medial habenula (dmHb) somas due to its rapid transport to axon terminal [30]. In the IPN, both SP[+] fibers, from the dmHb, as well as ChAT[+] fibers, from the vmHb, displayed strong GFP signal (Fig 1D). GFRα1 immunohistochemistry (Fig 1E) and in situ hybridization studies (S2 Fig) confirmed strong expression in the somas of adult mHb neurons and their axons throughout the IPN. To assess GFRα1 expression in the GABAergic IPN neurons, we used *R1CG*[LoxP] mice, expressing GFP from the *Gfra1* locus upon Cre-mediated recombination [31], bred to *Gad67*[Cre] mice, which express Cre in GABAergic neurons [32]. This revealed a small and sparse population of GFP[+] cells in the dorsal IPN, which did not appear labeled for GFRα1 immunohistochemistry (Fig 1F), possibly reflecting GFP that persisted after transient expression of GFRα1 during development. At any rate, absence of a significant population of GFRα1 expressing cells in the IPN is in agreement with almost undetectable levels of *Gfra1* mRNA in this structure by real-time RT-PCR (S1B Fig; see also S3H Fig) or by in situ hybridization

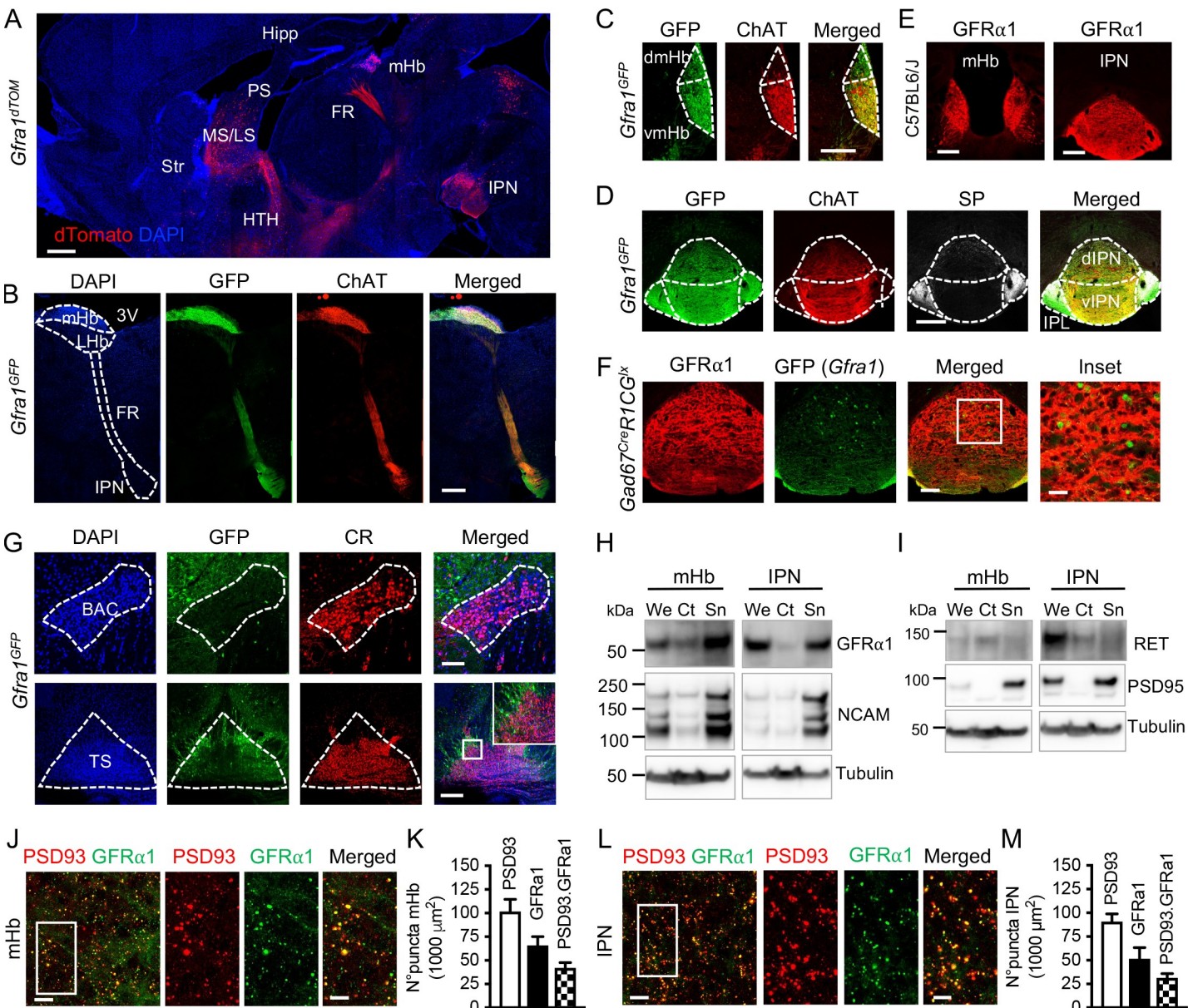

**Fig 1. Characterization of GFRα1 and co-receptor expression in neurons and synapses of the septum→Hb→IPN pathway. (A)** dTomato epifluorescence (red) in sagittal sections of *Gfra1^dTOM^* mouse brain injected with tamoxifen at 3 months counterstained with DAPI (blue). Scale bar, 500 μm. **(B)** GFP (green) and ChAT (red) immunostaining in sagittal sections of a 3-month-old *Gfra1^GFP^* mouse brain counterstained with DAPI (blue). Scale bar, 400 μm. **(C, D)** GFP (green), ChAT (red), and SP (gray) immunostaining in coronal sections of the mHb (C) and IPN (D) of a 3-month-old *Gfra1^GFP^* mouse brain. Scale bars, 200 μm. **(E)** GFRα1 immunostaining (red) in coronal sections of the mHb and IPN from a 3-month-old control (C57BL6/J) mouse. Scale bars, 200 μm. **(F)** GFP (green) and GFRα1 (red) immunostaining in a coronal section containing the IPN of a 3-month-old *Gad67^Cre^;R1CG^fx/fx^* mouse. GFP signal represents the GABAergic cells that express *Gfra1*. Scale bars, 100 μm (merged) and 35 μm (inset). **(G)** GFP (green) and CR (red) immunostaining in coronal sections of the BAC and TS from a 3-month-old *Gfra1^GFP^* mouse brain counterstained with DAPI (blue). Scale bars, 100 μm (BAC) and 200 μm (TS). **(H, I)** Immunoblots of We, Ct, and Sn protein extracts from the mHb and IPN of 3-month-old C57BL6/J mice probed for GFRα1 and NCAM (H) or RET and PSD95 (I). Tubulin was probed as loading control. **(J, M)** GFRα1 (green) and PSD93 (red) immunostaining in the mHb (J) and IPN (L) of 3-month-old C57BL6/J mice. Scale bars, 10 μm (merged) and 5 μm (insets). Graphs show the quantification (± SEM) of GFRα1, PSD93, and double-labeled puncta in mHb (K) and IPN (M). *N* = 5 mice (25–30 images per mouse per structure). The data underlying this figure can be found at https://figshare.com/projects/Raw_Data_Fernandez-Suarez_et_al_2021/123406. 3V, 3th ventricle; ChAT, choline acetyltransferase; CR, calretinin; Ct, cytosolic protein fraction; dIPN, dorsal interpeduncular nucleus; dmHb, dorsal medial habenula; FR, fasciculus retroflexus; GFRα1, glial cell–derived neurotrophic factor receptor alpha 1; Hipp, hippocampus; HTH, hypothalamus; IPL, lateral interpeduncular nucleus; IPN, interpeduncular nucleus; LHb, lateral habenula; LS, lateral septum; mHb, medial habenula; MS, medial septum; NCAM, neural cell adhesion molecule; PSD, postsynaptic density; Sn, synaptosome protein fraction; SP, substance P; Str, striatum; vIPN, ventral interpeduncular nucleus; vmHb, ventral medial habenula; We, whole protein extract.

(S2 Fig). As for GFRα1 co-receptors, NCAM was abundant in both the LHb and mHb subnuclei, overlapping with the *Gfra1*dTOM signal in the latter (S1F Fig). By contrast, *RET*GFP signal was only detected in the LHb, but not in the mHb (S1F Fig). Axonal projections labeled by *Gfra1*dTOM and *RET*GFP also appeared segregated in the FR, in agreement with the peripheral localization of LHb axons [33] (S1F Fig). NCAM was detected throughout the IPN, while *RET*GFP signal was only observed in scattered cells localized to the dorsal portion without significant overlap with mHb axons labeled by *Gfra1*dTOM (S1G Fig). In the septum, no *Gfra1*GFP signal could be detected in calretinin (CR)⁺ neurons of the BAC or TS, the 2 subpopulations that project to the mHb (Fig 1G). To ascertain whether the remaining *Gfra1*GFP cells observed adjacent to the TS projected to the mHb, we performed fluorogold retrograde tracing by stereotaxic injection in the mHb, followed by examination of fluorogold co-localization with *Gfra1*GFP signal in the TS (S3A and S3B Fig). Fluorogold did not overlap with GFP in the TS, although it did with CR, as expected (S3C and S3D Fig), indicating that septal axons projecting to the mHb do not carry GFRα1. In the TS, a sparse cell subpopulation displayed *RET*GFP signal, but it did not overlap with either the fluorogold tracer injected in the mHb (S3E–S3H Fig), *Gfra1*dTOM signal (S3I Fig), or CR (S3G and S3H Fig). No *RET*GFP signal could be detected in BAC (S3G Fig). Finally, NCAM was readily detected in both TS and BAC, where it decorated the plasma membrane of CR⁺ projection neurons (S3J and S3K Fig).

Next, we assessed whether GFRα1 is located at synapses in the septum→mHb→IPN pathway. By western blotting, we established the enrichment of GFRα1 and NCAM in synaptosomal fractions purified from adult mHb and IPN whole tissue extracts, as identified by the synaptic marker postsynaptic density (PSD)95 (Fig 1H and 1I). RET could not be detected in mHb extracts and was excluded from synaptosomes in the IPN (Fig 1I). By immunohistochemistry, approximately two-thirds of all GFRα1⁺ puncta in the mHb and in the IPN co-localized with the synaptic marker PSD93 (Fig 1J–1M), confirming a predominantly synaptic location of GFRα1 in these structures. Although the resolution of this analysis does not allow to determine the subsynaptic localization of GFRα1, the absence of GFRα1 from septal neurons projecting to the mHb, and from IPN neurons, would indicate that GFRα1 is postsynaptic in septohabenular synapses and presynaptic in habenulointerpeduncular synapses.

## Adult ablation of GFRα1 induces presynaptic alterations and loss of glutamatergic synapses in the mHb and IPN

Global ablation of GFRα1 expression in adult mice was attained by crossing *Gfra1*LoxP/+ and *Gfra1*CreERT2/+ strains to obtain *Gfra1*+/+, *Gfra1*CreERT2/+, and *Gfra1*CreERT2/LoxP animals, herein referred to as wild type (WT), heterozygous (Het), and knockout (KO), respectively. Tamoxifen treatment at 3 months of age reduced GFRα1 expression to near background levels in the mHb and IPN of KO mice (S4A–S4H Fig). Specific ablation of GFRα1 expression in the mHb was achieved by bilateral, stereotaxic injection of adeno-associated viruses (AAV2) expressing CRE recombinase under the control of a CMV promoter into the mHb of 3-month-old *Gfra1*fx/fx mice (herein called mHb.KO). Virus injection reduced GFRα1 levels by 56% and 60% in mHb and IPN, respectively, compared to mock-injected animals (herein called mHb. WT) (S4I–S4K Fig).

The numbers of synaptic puncta double-labeled by pre- and postsynaptic markers synapsin 1 and PSD93, respectively, were decreased by approximately half in both mHb and IPN of KO mice compared to WT 28 days after tamoxifen treatment (Fig 2A and 2B), suggesting loss of synapses after global ablation of GFRα1. A very pronounced decrease in the number of glutamatergic puncta labeled by vesicular glutamate transporters 1 and 2 (VGlut1 and VGlut2) was also observed in the mHb and IPN of KO mice 28 days after tamoxifen administration (Fig 2C

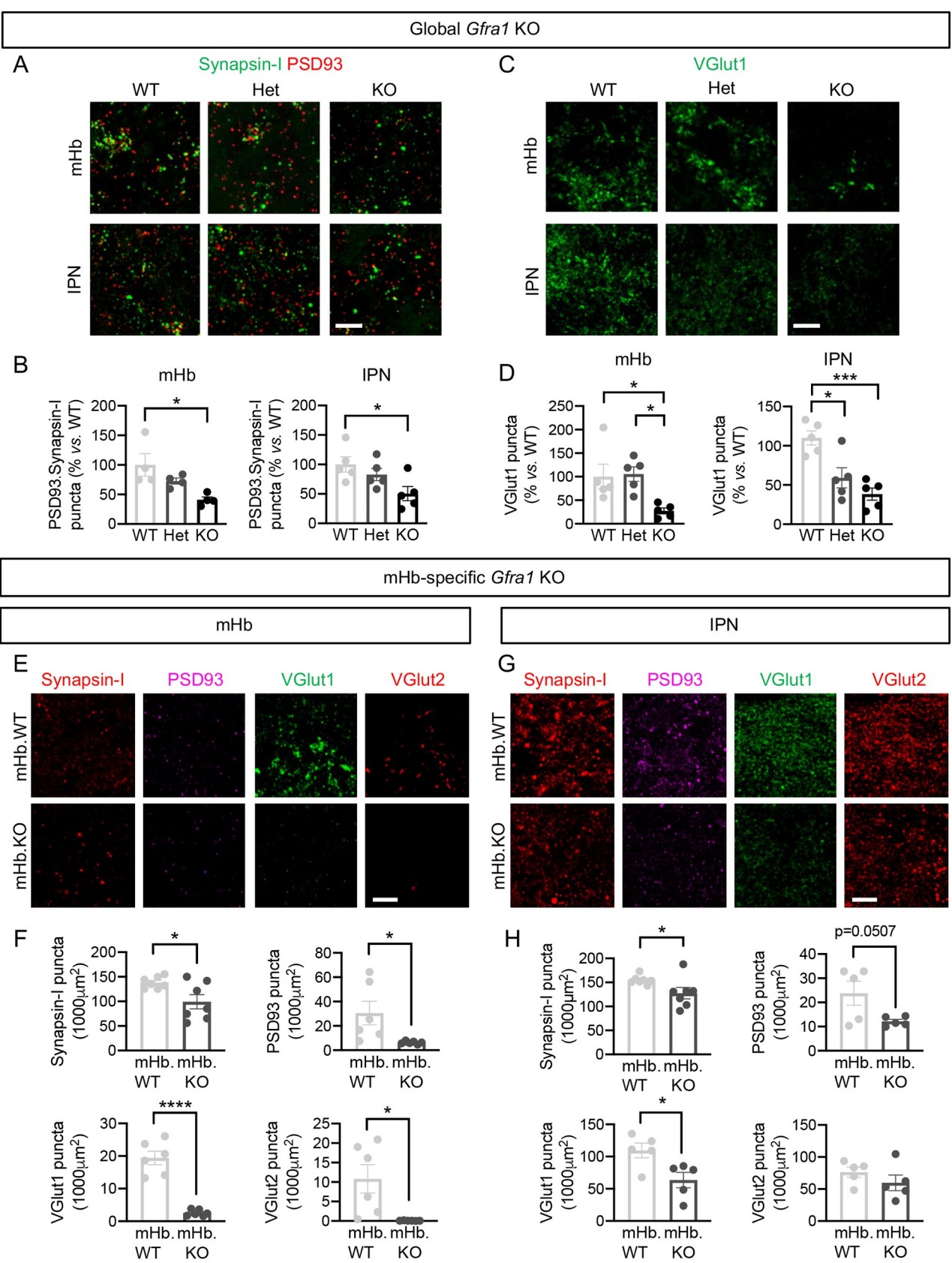

**Fig 2. Global and mHb-specific ablation of GFRα1 induces loss of glutamatergic synapses in mHb and IPN neurons. (A)** Synapsin-I (green) and PSD93 (red) immunostaining in coronal sections of the mHb and IPN of WT, Het, and KO mice sacrificed 1 month after tamoxifen treatment. Scale bar, 10 μm. **(B)** Quantification (± SEM) of puncta with co-localized immunoreactivity for Synapsin-I and PSD93 in mHb and IPN. $N$ = 4–5 mice per group (25–30 images per mouse per structure); 1-way ANOVA followed by Tukey post hoc test; *, $P < 0.05$. **(C)** VGlut1 (green) immunostaining in coronal sections of the mHb and IPN of WT, Het, and KO mice sacrificed 1 month after tamoxifen treatment. Scale bar, 10 μm. **(D)** Quantification (± SEM) of VGlut1 puncta in the mHb and IPN. $N$ = 5 mice per group (25–30 images per mouse per structure); 1-way ANOVA followed by Tukey post hoc test; *, $P < 0.05$; ***, $P < 0.001$. **(E, G)** Synapsin-I (red), PSD93 (purple), VGlut1 (green), and VGlut2 (red) immunostaining in coronal sections of the mHb (E) and IPN (G) from mHb.KO or mHb.WT mice sacrificed 1 month after viral delivery into the mHb. Scale bar, 10 μm. **(F, H)** Quantification (± SEM) of puncta with immunoreactivity for each marker in mHb (E) and IPN (G) of mHb.KO or mHb.WT mice. $N$ = 5–7 mice per group (25–30 images per mouse per structure); Student $t$ test; *, $P < 0.05$; ****, $P < 0.0001$. The data underlying this figure can be found at https://figshare.com/projects/Raw_Data_Fernandez-Suarez_et_al_2021/123406. GFRα1, glial cell–derived neurotrophic factor receptor alpha 1; Het, heterozygous; IPN, interpeduncular nucleus; KO, knockout; mHb, medial habenula; PSD93, postsynaptic density protein-93; VGlut1, vesicular glutamate transporter 1; WT, wild-type.

and 2D, S5A and S5B Fig). By contrast, mHb synaptic puncta labeled by vesicular GABA transporter (VGAT) and gephyrin, corresponding to inhibitory synapses, were not affected (S5C and S5D Fig), nor were IPN synaptic puncta labeled by vesicular acetyl choline transporter (VAChT) (S5E and S5F Fig). This indicated a specific loss of glutamatergic synapses in the mHb and the IPN of KO mice. Importantly, the numbers of synaptic puncta labeled by synapsin 1, PSD93, VGlut1, and VGlut2 were all significantly reduced in the mHb and IPN of mHb.KO mice 28 days after Cre virus injection (Fig 2E–2H), confirming the specific requirement of adult mHb GFRα1 expression for the maintenance of glutamatergic synapses in both the mHb and IPN.

Synapses in the mHb and IPN were further characterized by transmission electron microscopy. In agreement with our immunohistochemistry results, the number of synapses in the mHb was decreased by half in KO mice compared to Het or WT animals (Fig 3A and 3B). Intriguingly, a marked presynaptic phenotype was observed in the remaining synapses, characterized by a reduction in the number of presynaptic vesicles, which affected both KO and Het mice (Fig 3A and 3B). On the other hand, the postsynaptic length and the width of the synaptic cleft were not affected (Fig 3A and 3B). Similar defects were observed in S synapses in the IPN, previously shown to correspond to mHb afferents [34] (Fig 3C and 3D). We note that there were no differences in mHb neuron number between WT, Het, and KO mice 4 weeks after tamoxifen injection (S5G and S5H Fig), indicating no effects on mHb neuron survival after acute ablation of GFRα1 in adult mice. Together, these data indicate that adult expression of GFRα1 in the mHb is essential for the maintenance of both the number and structure of glutamatergic synapses in the mHb and IPN.

### Adult loss of GFRα1 affects the functional connectivity of the septum→Hb→IPN pathway and alters AMPAR-mediated glutamatergic responses in mHb neurons

In order to assess whether adult loss of GFRα1 affects connectivity in the septum→mHb→IPN pathway, we used a chemogenetic approach. In the first set of experiments (Fig 4A), an AAV2 vector containing a designer receptor exclusively activated by designer drug (DREADD) construct driving Cre-dependent expression of hM3Dq and mCherry was stereotaxically injected in the TS of WT, Het, and KO mice (Fig 4A). Since the TS neurons projecting to the mHb do not express GFRα1 (i.e., no Cre expression in Het or KO), a second AAV2 construct expressing both Cre and GFP was delivered together with the DREADD construct. Additional control experiments showed that all genotypes were infected by the AAV2 vectors with comparable efficiency in the TS, as no significant differences were observed in the number of GFP+ cells in the TS neither in the OD for mCherry in the terminals in the mHb (S6A–S6D Fig). Treatment with clozapine N-oxide (CNO) for 1 hour resulted in strong cFOS signals in the TS and mHb

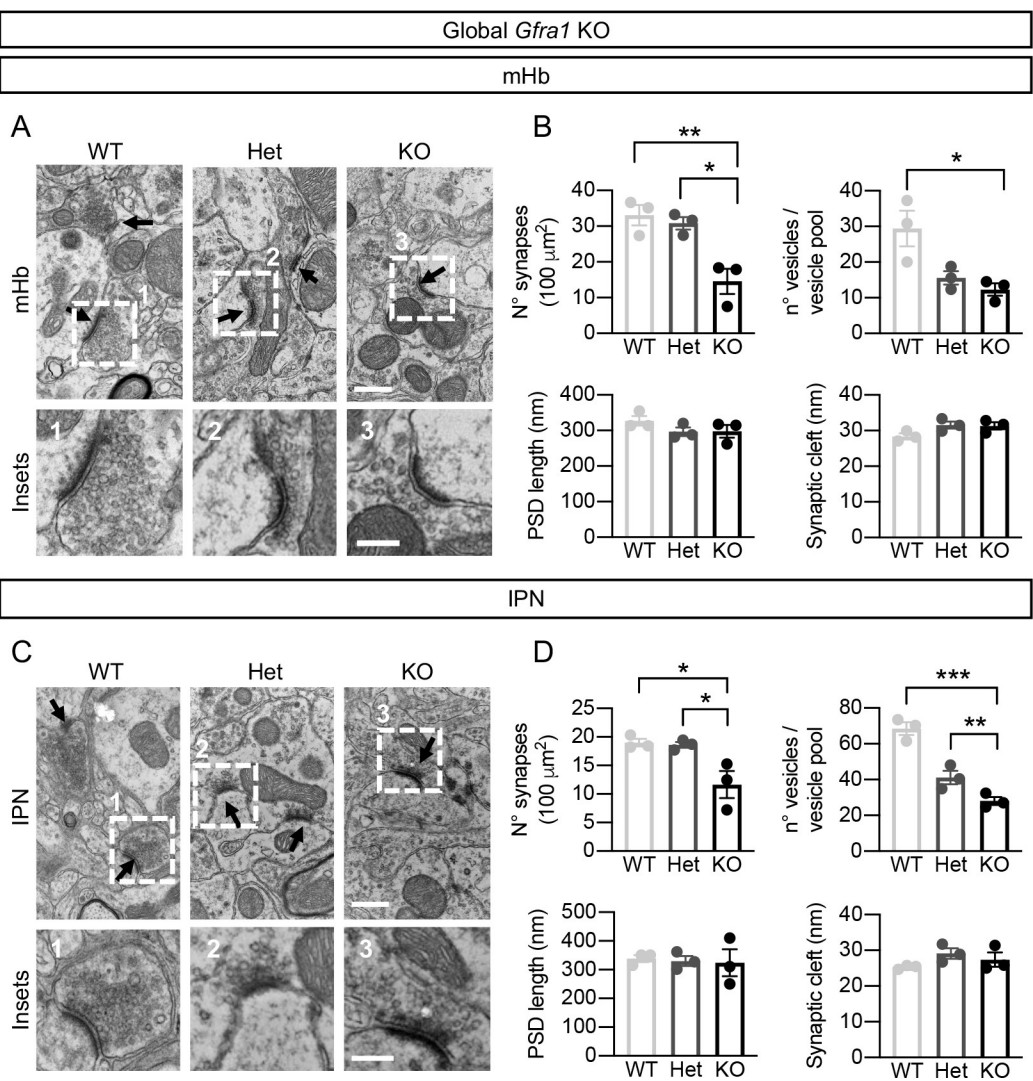

**Fig 3. Reduced number of synapses and presynaptic vesicles in mHb and IPN after acute ablation of GFRα1 in adult mice. (A, C)** TEM images showing asymmetric synapses in the mHb (A) and S-synapses in the IPN (C) of WT, Het, and KO mice sacrificed 1 month after tamoxifen treatment. Scale bars, 500 nm and 200 nm (insets). **(B, D)** Quantification (± SEM) of synapse density, synaptic cleft width, number of presynaptic vesicles and PSD length in mHb (B; 50 images per mouse, a total of 240, 207, and 115 synapses analyzed per genotype) and IPN (D; 30–50 images per mouse, 216, 157 and 104 S-synapses analyzed per genotype) of WT, Het, and KO mice. $N = 3$ mice per group; 1-way ANOVA followed by Tukey post hoc test; *, $P < 0.05$; **, $P < 0.01$; ***, $P < 0.001$). The data underlying this figure can be found at https://figshare.com/projects/Raw_Data_Fernandez-Suarez_et_al_2021/123406. GFRα1, glial cell–derived neurotrophic factor receptor alpha 1; HET, heterozygous; IPN, interpeduncular nucleus; KO, knockout; mHb, medial habenula; PSD, postsynaptic density; TEM, transmission electron microscopy; WT, wild-type.

only when the DREADD construct was expressed by the septal neurons (Fig 4B, S6 Fig). Importantly, while there were no differences in the number of cFOS+ cells in the TS after CNO treatment (S6A and S6C Fig), the number of cFOS+ cells in the mHb was significantly reduced in both Het and KO compared to WT (Fig 4C), indicating that GFRα1 is required to maintain functional connectivity in the septum-mHb pathway.

In the second set of experiments, the same DREADD virus was injected in the mHb, while a second AAV2 virus expressing only GFP was used as control. In this case, only Het and KO mice express Cre (from the *Gfra1* locus); thus, in WT mice, no hM3Dq expression could be

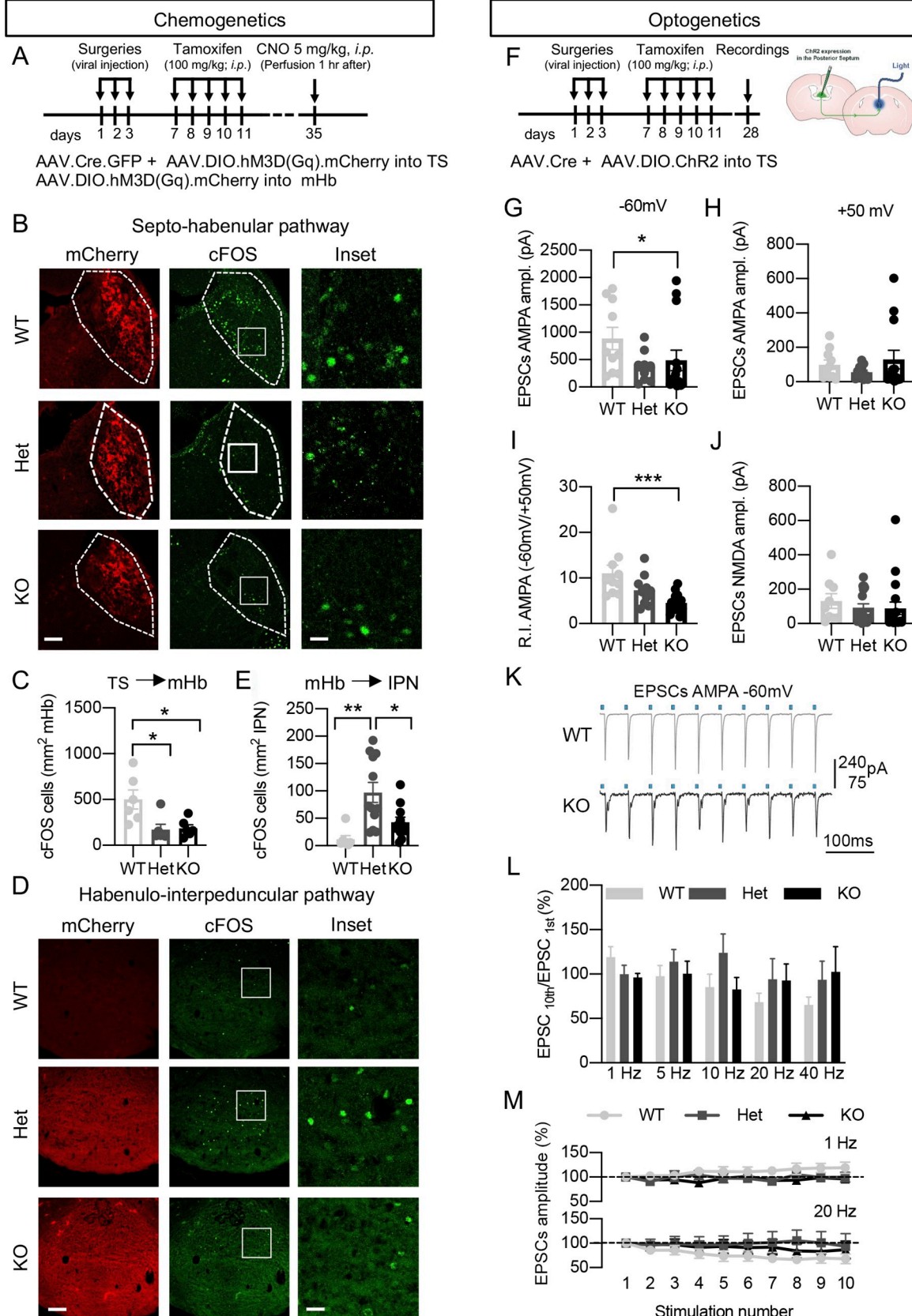

**Fig 4. Adult loss of GFRα1 affects the functional connectivity of the septum→Hb→IPN pathway and alters AMPAR-mediated glutamatergic responses in mHb neurons. (A)** Schematics of the chemogenetic procedures. WT, Het, and KO mice were injected with (i) a combination of AAV.Cre and AAV.hM3D(Gq) viruses into the TS or with (ii) the DREADD virus in the mHb and treated with tamoxifen 1 week later. All animals were sacrificed 28 days later 1 hour after CNO treatment. **(B, C)** mCherry epifluorescence (red) and cFOS (in green) immunostaining in the mHb of WT, Het, and KO mice after chemogenetic activation of TS projecting neurons expressing hM3Dq (B). Scale bars, 70 μm and 15 μm (insets). Graph (C) shows the quantification (± SEM) of cFOS positive cells in mHb of WT, Het, and KO mice ($N$ = 6 animals/group, 10–14 images per mouse); 1-way ANOVA followed by Tukey multiple comparison test; *, $P < 0.05$. **(D, E)** mCherry epifluorescence (red) and cFOS (in green) immunostaining in the IPN of WT, Het, and KO mice after chemogenetic activation of mHb neurons expressing hM3Dq (D). Scale bars, 100 μm and 25 μm (insets). Graph (E) shows the quantification (± SEM) of cFOS positive cells in the IPN of WT ($N$ = 7), Het ($N$ = 12), and KO ($N$ = 11) mice (5–6 images per mouse); 1-way ANOVA followed by Tukey multiple comparison test; * $P < 0.05$; ** $P < 0.01$. **(F)** Schematics of the optogenetics procedures. WT, Het, and KO mice were injected with a combination of AAV.Cre and AAV.ChR2 viruses into the TS and treated with tamoxifen 1 week later. Three weeks after, brain sections were obtained, and the response of the mHb neurons was recorded after optogenetic activation of the septal terminals. **(G–I)** EPSC average amplitudes at −60 mV (G) and +50 mV (H) mediated by AMPA receptors after optogenetic stimulation of septal fibers recorded in the presence of NMDA antagonist APV. R.I. (I) was calculated as the ratio between the EPSC amplitudes at −60 mV and +50 mV. $N$ = 10–14 cells per genotype; Kruskal–Wallis followed by Dunn multiple comparison test; *, $P < 0.05$; ***, $P < 0.001$. **(J)** NMDAR-mediated EPSCs at +50 mV in WT ($n$ = 9), Het ($n$ = 17), and KO ($n$ = 18) mHb cells after optogenetic stimulation of the septal terminals. **(K)** Representative AMPAR-mediated EPSCs recorded at −60 mV in mHb of WT and KO mice during 10 consecutive optogenetic 20-Hz stimulations. **(L, M)** EPSC amplitude (± SEM) (L) and its variation (M) between first and 10th stimulation at different frequencies of WT, Het, and KO mice ($n$ = 10–21 cells per frequency per genotype). The data underlying this figure can be found at https://figshare.com/projects/Raw_Data_Fernandez-Suarez_et_al_2021/123406. DREADD, designer receptor exclusively activated by designer drug; EPSC, excitatory postsynaptic current; GFRα1, glial cell–derived neurotrophic factor receptor alpha 1; Het, heterozygous; IPN, interpeduncular nucleus; KO, knockout; mHb, medial habenula; R.I., rectification index; TS, triangular septum; WT, wild-type.

detected in the mHb (S7A and S7B Fig) neither cFOS signals in the mHb (S7A and S7C Fig) or the IPN (Fig 4D) after CNO treatment. cFOS induction could neither be detected in WT, Het, or KO mice injected with the GFP virus after CNO treatment (S7D–S7F Fig), nor after delivery of the DREADD virus when tamoxifen (S7G Fig) or CNO were omitted (S7H Fig). Importantly, the efficiency of transfection with the DREADD virus was very similar between Het and KO mice, as reflected in the OD for mCherry and the number of cFOS[+] cells in the mHb after CNO treatment (S7A–S7C Fig). Treatment with CNO induced strong cFOS signals in the IPN only in the presence of the DREADD virus (Fig 4D, S7D Fig). In this case, the number of cFOS[+] cells was significantly reduced in the IPN of KO mice compared to Het mice (Fig 4E), indicating impaired connectivity in the adult mHb-IPN pathway after acute loss of GFRα1.

The cFOS responses observed in the mHb reflected the result of sustained chemogenetic stimulation (1-hour CNO treatment) of TS neurons. The decrease observed in Het and KO mice is in line with their reduced number of presynaptic vesicles compared to WT animals, as a smaller vesicle pool would be faster depleted following prolonged activation. In order to investigate faster synaptic responses, we performed electrophysiological measurements of AMPAR- and NMDAR-mediated components in the mHb after optogenetic stimulation of TS afferents. AAV2 vectors expressing Cre-dependent Channelrhodopsin-2 (ChR2) or Cre recombinase were stereotaxically co-injected in the TS of WT, Het, and KO mice (Fig 4F). Brain slices containing the mHb were prepared from 3 weeks after tamoxifen treatment, and excitatory postsynaptic currents (EPSCs) elicited by optogenetic stimulation of TS terminals were recorded in voltage-clamped mHb neurons. These EPSCs could be completely blocked by the AMPAR and NMDAR antagonists NBQX and D-APV, respectively, indicating that they were purely glutamatergic, in agreement with previous studies [12]. We found that the amplitude of AMPAR-mediated EPSCs recorded at −60 mV in the presence of the NMDAR antagonist D-APV was significantly reduced in mHb neurons of KO mice compared to WT (Fig 4G). On the other hand, no difference between genotypes could be found in EPSCs recorded at +50 mV (Fig 4H). As a result, the ratio between EPSCs amplitudes measured at these 2 voltages, which constitutes the rectification index (R.I.) of the AMPAR component of these EPSCs, was significantly lower in mHb KO neurons compared to WT neurons (Fig 4I).

Neither AMPAR-mediated EPSCs amplitude nor R.I. of Het neurons were significantly different from WT (Fig 4G–4I). Importantly, NMDAR-mediated EPSCs were not altered in KO mHb neurons (Fig 4J). Neither the dynamic behavior of the synapses in response to trains of 10 stimulations at frequencies ranging from 1 to 40 Hz was different between KO and WT mHb neurons (Fig 4K–4M), suggesting that the differences observed between genotypes in EPSCs amplitude were due to postsynaptic alterations. We note that the presence of rectifying AMPAR-mediated EPSCs (Fig 4I) is evidence of the existence of $Ca^{2+}$-permeable AMPARs in mHb neurons, in agreement with previous observations [12]. Thus, the lower R.I. value of AMPAR-mediated EPSCs in KO mHb neurons indicated reduced $Ca^{2+}$ permeability of AMPAR complexes in mHb neurons upon removal of GFRα1, suggesting possible changes in the composition of postsynaptic AMPAR subunits in these neurons.

## Altered molecular composition of AMPAR complexes in mHb neurons after adult ablation of GFRα1

AMPAR are tetrameric ion channels formed by dimers of dimers derived from any of 4 different subunits, termed GluA1 to 4, in different combinations [35]. In the adult brain, the majority of AMPAR complexes contain GluA2 subunits that have undergone RNA editing, making the assembled channel $Ca^{+2}$ impermeable [36]. On the other hand, AMPAR lacking GluA2 are $Ca^{+2}$-permeable and, although very abundant in the developing CNS, they are not common in the adult [35]. Although earlier electrophysiological data have indicated the presence of $Ca^{2+}$-permeable AMPAR complexes in the septohabenular–interpeduncular pathway [9,12], the molecular composition and functional importance of these complexes are unknown. Immuno-histochemistry for each of the 4 GluA subunits in the mHb of adult WT mice revealed high levels of GluA1 and GluA4, lower expression of GluA2, and negligible expression of GluA3 (S8A Fig). In the IPN, all GluA subunits were present, with surprisingly high levels of GluA4 (S8A Fig). Western blot analysis of whole tissue, cytosolic, and synaptic fractions of the mHb and IPN revealed the presence of all 4 subunits in both structures (S8B Fig). While the GluA3 blot had to be overexposed to detect specific signals, GluA4 was again very highly expressed in the IPN (S8B Fig). To better understand the composition of AMPAR complexes in the mHb and IPN, the proximity ligation assay (PLA) was used to investigate interactions in situ between GluA1 and GluA2 subunits, the most common $Ca^{+2}$-impermeable channels in the brain, and between GluA1 and GluA4, forming $Ca^{+2}$-permeable complexes. Both types of complexes were equally present in the dorsal and the ventral portions of the mHb (S8C and S8D Fig), while in the IPN, the lateral subnuclei presented less GluA1/GluA2 and GluA1/GluA4 complexes than the core IPN (S8C and S8E Fig).

Having established the presence of both GluA1/GluA2 ($Ca^{2+}$ impermeable) as well as GluA1/GluA4 ($Ca^{2+}$ permeable) AMPAR complexes in mHb and IPN, we examined possible changes in the levels, arrangement, and phosphorylation state of these subunits upon removal of GFRα1, as any of these can lead to changes in synaptic strength and function [37]. In the mHb, the level of GluA2 was significantly increased in KO mice (Fig 5A), while in the IPN, both GluA1 and GluA4 levels were reduced in mice lacking GFRα1 (Fig 5B). No statistically significant changes were observed in Het mice. Whereas GluA1 phosphorylation at $Ser^{831}$ was unchanged in the mutants, phosphorylation at $Ser^{845}$ was reduced in the mHb of KO mice (Fig 5C–5F). Phosphorylation at $Ser^{845}$ has been implicated in the insertion of GluA1 into synapses [38]. Changes in the composition of AMPAR complexes were evaluated by PLA. GluA1/GluA2 interactions were found to be significantly increased in the mHb and core IPN of KO mice (Fig 5G and 5H), while GluA1/GluA4 complexes were decreased in the core IPN of the mutants (Fig 5I and 5J), indicating that adult loss of GFRα1 leads to rearrangements in the

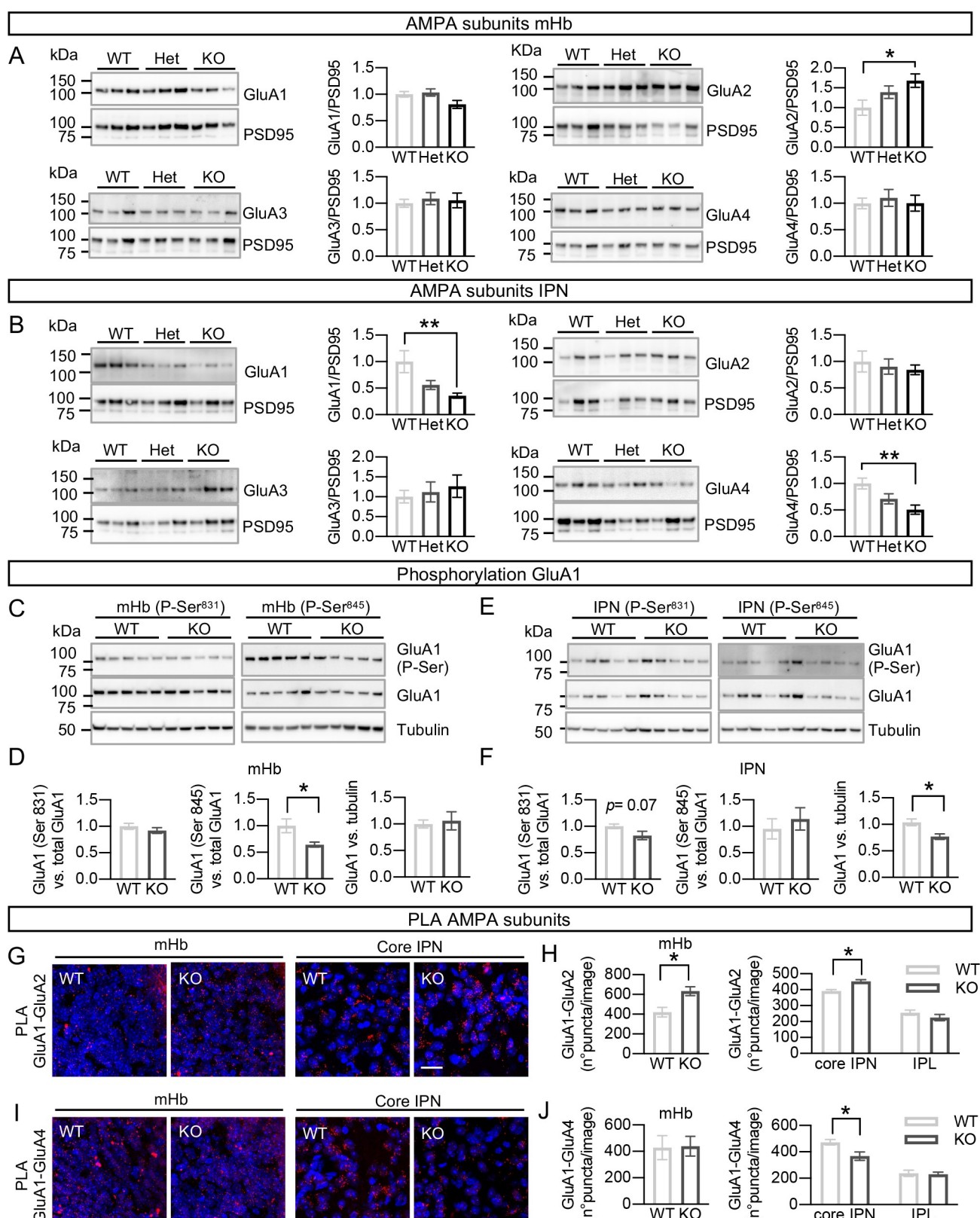

**Fig 5. Altered molecular composition of AMPAR complexes in mHb neurons after adult ablation of GFRα1. (A, B)** Immunoblots of synaptic protein fractions from mHb (A) and IPN (B) of WT, Het, and KO mice sacrificed 1 month after tamoxifen treatment probed for GluA1, 2, 3 and 4 as

indicated. PSD95 was probed as loading control. Quantifications (± SEM) of GluA1 to 4 levels were corrected for PSD95 levels and normalized to levels in WT samples. $N$ = 9–10 samples per group; 1-way ANOVA analysis followed by Tukey multiple comparison test; * $P < 0.05$; ** $P < 0.01$. **(C, E)** Immunoblots of synaptic protein fractions from mHb (C) and IPN (E) of WT, Het, and KO mice probed for GluA1 P-Ser[831] and P-Ser[845] as indicated and reported for total GluA1 and Tubulin as loading controls. **(D, F)** Quantifications (± SEM) of GluA1 P-Ser[831] and P-Ser[845] levels in mHb (D, $N$ = 13 samples per group) and IPN (F, $N$ = 14 samples per group) corrected for total GluA1 levels and normalized to levels in WT samples. Student $t$ test; * $P < 0.05$. **(G, I)** PLA signals (red) for GluA1-GluA2 (G) and GluA1-GluA4 (I) complexes in coronal sections of mHb and core IPN from WT and KO mice. Counterstaining with DAPI appears in blue. Scale bars, 20 μm. **(H, J)** Quantification (± SEM) of PLA puncta for GluA1-GluA2 (H) and GluA1-GluA4 (J) complexes in mHb, core IPN and lateral IPN (IPL) from WT and KO mice ($N$ = 5 mice per group). A total of 16 images from the mHb (8 dorsal and 8 ventral) and 18 images from the IPN (core IPN: 6 dorsal and 6 ventral; 6 lateral) were analyzed per mouse. Student $t$ test; * $P < 0.05$. The data underlying this figure can be found at https://figshare.com/projects/Raw_Data_Fernandez-Suarez_et_al_2021/123406. GFRα1, glial cell–derived neurotrophic factor receptor alpha 1; HET, heterozygous; IPN, interpeduncular nucleus; KO, knockout; mHb, medial habenula; PLA, proximity ligation assay; PSD, postsynaptic density; WT, wild-type.

composition of postsynaptic AMPAR subunits in the septohabenular–interpeduncular pathway, which are consistent with the reduced $Ca^{2+}$ permeability of those channels.

### Acute loss of GFRα1 in adult mHb neurons reduces anxiety-like behavior and potentiates context-based fear responses

Selective lesion of the TS and BAC projections to the mHb has been shown to increase fear responses to electric shock while decreasing anxiety-like behavior [4]. We performed a battery of behavioral tests 2 weeks after tamoxifen administration in WT, Het, and KO mice or 2 weeks after Cre virus injection in mHb.KO mice. The open field and elevated plus maze were used to evaluate anxiety-like behavior; foot shock-induced freezing behavior, to evaluate innate fear responses; and passive avoidance and fear conditioning, to evaluate fear-based learning. General locomotor activity in the open field was unchanged among genotypes (S9A Fig). In the elevated plus maze, KO mice entered more times and spent significantly longer time in the open arms than WT and Het animals (Fig 6A), suggesting reduced anxiety-like behavior in adult mice lacking GFRα1. Importantly, similar results were obtained in mHb.KO mice after Cre virus injection to the mHb (Fig 6B and 6D, S9B and S9D Fig), indicating that GFRα1 is required in the adult mHb for normal anxiety-like behavior.

Freezing responses after 3 consecutive foot shocks were similar between WT, Het, and KO mice as well as mHb.KO and mHb.WT animals (S9G Fig), indicating unaltered innate fear-induced freezing behavior in this test after loss of GFRα1. In the passive avoidance test (Fig 6C), both global (KO) and mHb-specific (mHb.KO) ablation of GFRα1 significantly increased the avoidance latency (Fig 6D and 6E), indicating potentiated responses to fear-based learning after acute loss of habenular GFRα1. To assess fear conditioning, mice were trained to pair a foot shock with either a specific context (Fig 6F) or a tone cue (S9I Fig). In the context fear conditioning, there were no differences between the WT, Het, and KO groups (Fig 6G, S9H Fig), but mHb.KO mice lacking GFRα1 only in mHb neurons showed an exacerbated freezing response after conditioning compared to mHb.WT controls (Fig 6H, S9H Fig). Finally, cued conditioned fear responses did not differ between WT, Het, and KO mice or between mHb. KO and mHb.WT mice (S9J and S9K Fig). Together, the results of these studies show that context-conditioned fear responses are significantly enhanced after acute loss of GFRα1 in adult mHb neurons.

## Discussion

Fear is an evolutionary conserved, adaptive behavior that helps organisms to react to threatening or otherwise dangerous situations [39]. On the other hand, persistent or exaggerated fear in the absence of danger is detrimental and maladaptive [40]. Previous research has shown that neurons in the mHb play a critical role in calibrating behavioral responses to threatening

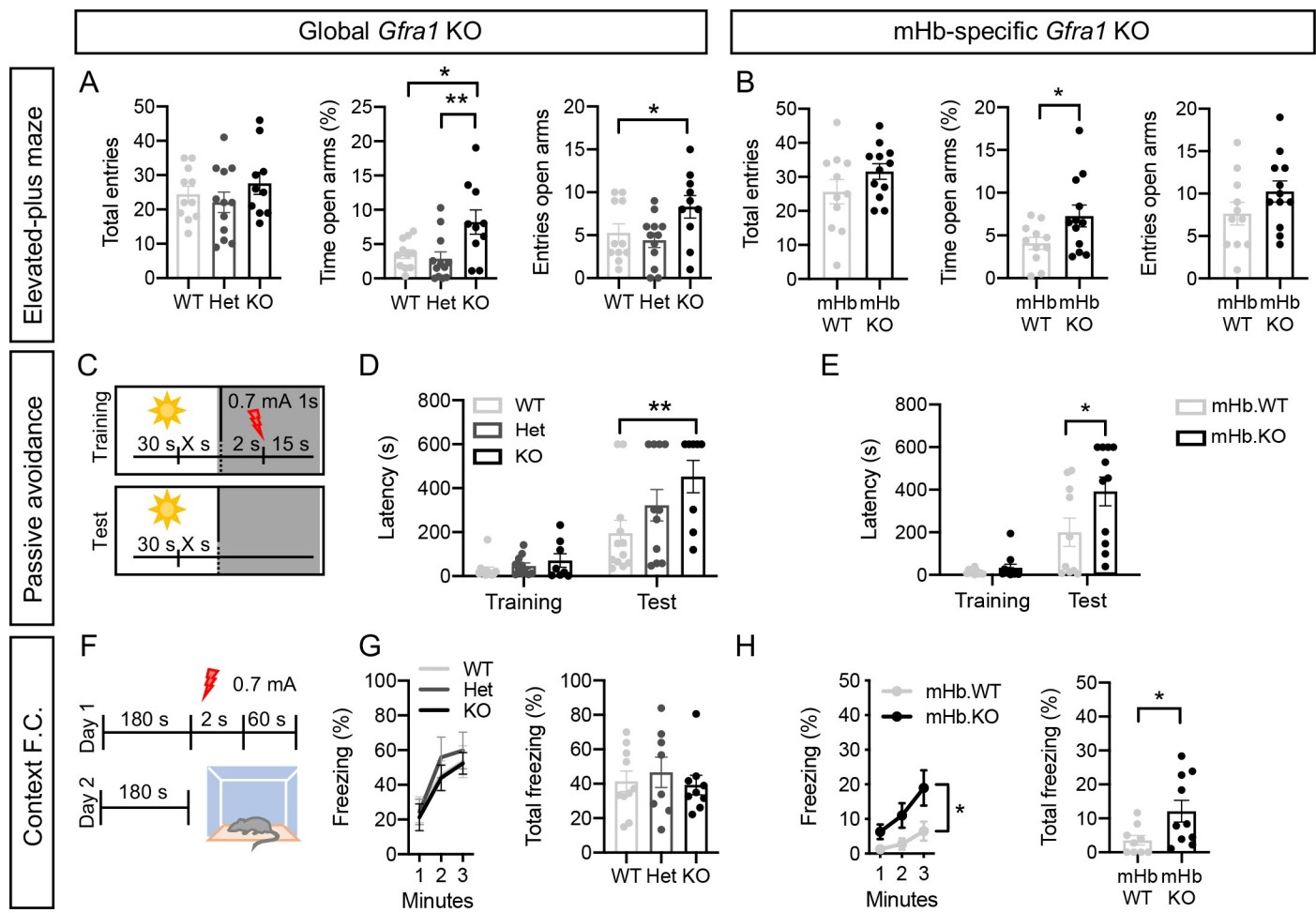

**Fig 6. Acute loss of GFRα1 in adult mHb neurons reduces anxiety-like behavior and potentiates context-based fear responses. (A, B)** Total entries, time spent in open arms and open arms entries (± SEM) for WT, Het, and KO mice (A) and mHb.WT and mHb.KO mice (B) in the elevated plus maze test. $N$ = 10–12 mice per group; 1-way ANOVA followed by Bonferroni post hoc test (A) and Student $t$ test (B); * $P$ < 0.05; ** $P$ < 0.01. **(C–E)** Schematic (C) and latencies (± SEM) during training and test for WT, Het, and KO mice (D) and mHb.WT and mHb.KO mice (E) in the passive avoidance test. $N$ = 8–12 mice per group; 2-way ANOVA followed by Bonferroni post hoc test; * $P$ < 0.05; ** $P$ < 0.01. **(F–H)** Schematic (F), freezing timecourse and total freezing time (± SEM) for WT, Het, and KO mice (G) and mHb.WT and mHb.KO mice (H) in the context fear conditioning test. $N$ = 8–10 mice per group; 1-way ANOVA followed by Tukey post hoc test (G) and Student $t$ test (H); *, $P$ < 0.05; **, $P$ < 0.01. The data underlying this figure can be found at https://figshare.com/projects/Raw_Data_Fernandez-Suarez_et_al_2021/123406. HET, heterozygous; KO, knockout; mHb, medial habenula; WT, wild-type.

situations in an experience-dependent manner [4,5]. MHb neurons display a number of features that set them apart from the majority of neurons in the adult brain, including excitatory GABAergic inputs and $Ca^{2+}$-permeable AMPA receptors [8,12]. However, the mechanisms that maintain and regulate the unique properties of mHb neurons and their function in mediating fear responses remain incompletely understood. Here, we demonstrate that the neurotrophic receptor GFRα1 regulates synapse homeostasis in adult mHb neurons to facilitate the normal processing of fear behavior. We report 3 major findings. First, GFRα1 is highly abundant in adult mHb neurons and their terminals, where it is localized to synapses, and required for the maintenance, structural integrity, and function of septohabenular and habenulointerpeduncular synaptic connections. Second, GFRα1 favors the expression of $Ca^{2+}$-permeable AMPA receptor complexes formed by the association of GluA1 and GluA4 subunits, at the expense of $Ca^{2+}$ impermeable GluA1/GluA2 complexes, in adult mHb synapses. Third, acute

loss of GFRα1 in adult mHb neurons reduces anxiety-like behavior and potentiates context-based fear responses, phenocopying the effects of lesions to mHb afferents from the posterior septum. These findings establish an unexpected role for GFRα1 as a critical mediator of the stability and function of glutamatergic synapses in the adult brain.

GDNF was originally discovered as a survival factor for midbrain dopaminergic neurons and a candidate anti-Parkinson disease agent [41], thereafter focusing research on GDNF receptors on the SNpc and VTA areas that harbor these cells. More recent studies characterized other sites of GFRα1 expression in the mammalian brain, particularly in GABAergic neurons of the cerebral cortex [42,43], cerebellum [44,45], and olfactory bulb [46,47]. Our present findings establish mHb neurons as the cells that express the highest levels of GFRα1 in the adult mouse brain, higher than any other known site of expression, and the first glutamatergic neurons known to require this receptor for appropriate connectivity and function. Although extensively studied in the developing nervous system, the functions of neurotrophic systems in the adult brain are less well understood. With the possible exception of neural injury and neurodegenerative disease, adult deletion of neurotrophic factors or their receptors has, in most cases, no effect on neuron maintenance. Our discovery of the requirement of GFRα1 for the structural and functional stability of mHb glutamatergic synapses represents the first demonstration of a function for this receptor in mature adult neurons. Despite being only expressed by mHb neurons, GFRα1 was found enriched at both septohabenular and habenulointerpeduncular synapses, suggesting both pre- and postsynaptic functions. This may be possible through the release of GFRα1 from the cell surface [15,16], facilitating non–cell-autonomous effects across the synapse and perisynaptic areas. In contrast to RET, expression of NCAM was detected in neurons along the septum→Hb→IPN pathway, where it was also concentrated at synapses, making it a possible candidate for mediating the effects of GFRα1 in this circuit. NCAM has been found to contribute to several functions in synapse development and maintenance beyond cell adhesion, including spine formation [48], synapse maturation [49], presynaptic vesicle recycling [50], and NMDA- and AMPA-mediated glutamatergic signaling [51]. GDNF, the main ligand of GFRα1, could also be detected in the septum, Hb, and IPN, although its specific cellular source, e.g., neurons or glia, remains to be established. The specific contributions, if any, of NCAM and GDNF to the function of GFRα1 in the septum→Hb→IPN pathway will require further investigation. On the other hand, the abundance of GFRα1 protein in cell bodies and terminals of mHb neurons suggests that other mechanisms may also be possible, including structural roles in overall synapse stability, and regulation of trafficking or turnover of AMPAR subunits. We also note that Het mice displayed an intermediate effect between WT and KO in several of the parameters evaluated, for example, in synaptic puncta, cFOS induction, and fear responses. Although GFRα1 haploinsufficiency during development cannot be ruled out, the fact that acute stereotaxic deletion in the adult mHb using an AAV.Cre virus, which also reduced GFRα1 expression by half (S4 Fig), produced effects that were quantitatively similar to those observed in KO mice (Figs 2 and 6) suggests that some of the effects observed in Het mice were caused by adult haploinsufficiency.

Our behavioral studies in adult mice following acute loss of GFRα1 in mHb neurons revealed an unexpected requirement for this receptor in normal anxiety-like behavior and conditioned fear responses. The behavioral effects observed in the mutants resembled lesions to glutamatergic TS and BAC afferents to the mHb [4], indicating that GFRα1 is a positive regulator of glutamatergic transmission in the septum→Hb→IPN pathway, and an essential component for normal mHb neuron function. The haploinsufficiency observed in Het mice for some of the phenotypes indicates a dosage effect and suggests a potential mechanism for adjusting the functional response of the pathway to dangerous or threatening situations by regulating the levels of GFRα1, its ligands, or co-receptors. In this context, it would be important

to identify the environmental cues and/or internal states that may affect GFRα1 expression in mHb neurons, as well as its trafficking and accumulation in mHb terminals. Finally, our findings open up a new avenue for investigations into the functional plasticity of mHb neurons, which will not only improve our understanding of psychiatric disorders associated with fear and anxiety, but also potentially lead to novel treatments for such disorders.

## Methods

### Mice

Mice were housed 2 to 5 per cage in standard conditions in a temperature and humidity-controlled environment, on a 12/12-hour light/dark cycle with ad libitum access to food and water. The following transgenic mouse lines were used for experiments: $EIIa^{Cre}$ [52], $Gad67^{Cre}$ [32], $Gfra1^{Cre-ERT2}$ [44], $Gfra1^{LoxP}$ (kindly provided by M. Saarma and J.O Andressoo, University of Helsinki), $R1CG^{LoxP}$, which expresses GFP from the $gfra1$ locus after Cre-mediated recombination [31], $Ret^{GFP}$ [29], and $Rosa26^{dTom}$ [53]. The $gfra1^{gfp}$ mouse line was generated by crossing $EIIa^{Cre}$ with $R1CG^{LoxP}$ mice to obtain GFP expression under the $Gfra1$ promoter in the germ cells and subsequently crossing the offspring with C57BL/6J mice. All animals were bred in a C57BL6/J background (Charles River, Netherlands) with the exception of the $gfra1^{LoxP}$ that was kept in a CD1 background (Charles River, Netherlands). Therefore, all animals resulting from breeding $Gfra1^{Cre-ERT2}$ and $Gfra1^{LoxP}$ mice were 50:50 C57BL6/J: CD1 background. Both males and females were used for these studies. Animal protocols were approved by Stockholms Norra Djurförsöksetiska Nämnd and are in accordance with the ethical guidelines of Karolinska Institutet (ethical permits N173-15; 11563–2018 and 11571–2018 following EU directive 2010/63/EU).

### Tamoxifen treatment and stereotaxic surgery

Adult mice (3 to 4 months old) were injected intraperitoneally (i.p.) with 100 mg/kg of tamoxifen (Sigma-Aldrich) dissolved in 10% EtOH in corn oil (Sigma-Aldrich) for 5 consecutive days. In the experiments with $Gfra1^{+/+}$, $Gfra1^{CreERT2/+}$, and $Gfra1^{CreERT2/LoxP}$ (WT, Het, and KO mice), all animals were injected with tamoxifen in all cases.

Adult mice (3 to 4 months old) were anesthetized with isoflurane (1% to 5%) and placed in a stereotaxic frame (David Kopf Instruments, CA, USA). Injection was performed with a wiretrol capillary micropipette (Drummond Scientific, PA, USA) by nanoliter pressure injection at a flow rate of 50 nL per min (Micro4 controller, Nanojector II, World Precision Instruments, FL, USA). The pipette was left on place for 10 minutes after the injection before retracting it slowly from the brain. $Gfra1^{LoxP}$ mice were injected bilaterally with 300 nL of AAV2-CMV-PI-Cre-rBG (Penn Vector Core, $2.27 \times 10^{13}$ genomic copies per mL) or vehicle in the following coordinates obtained from the Paxinos and Franklin atlas: (i) mHb (10˚ angle at the coordinates anteroposterior [AP] −1.94 mm, mediolateral [ML] ±0.75 mm, and dorsoventral [DV] −2.3 mm). Due to the lack of a reporter gene in the viral construct, the expression levels of GFRα1 in the mHb were evaluated in all cases by immunofluorescence at the end of the experiment and the ones showing no loss were removed from the experimental data set. To evaluate the transfection efficiency of the viral vector, $R1CG^{LoxP}$ mice were injected with the virus in the same coordinates, observing GFP expression under the $gfra1$ promoter after Cre recombination.

### Biochemical techniques

**Tissue processing.** Animals were sacrificed by cervical dislocation, brains were rapidly removed, dissected manually, frozen in dry ice, and stored at −80˚C until further use. For the

dissection of the mHb and the IPN, brains were cut coronally with a blade at −1 mm from bregma and +1 mm from lambda, obtaining a coronal section containing the habenula. The cortex and the hippocampus from that section were removed, leaving the mHb on sight, which was dissected by pulling it out carefully. Next the IPN, which could be visualized directly in the remaining caudal part of the brain, was collected.

**RNA extraction and real-time RT-PCR.** The mRNA levels of *Gfra1* and *Gdnf* were quantified by real-time RT-PCR in total RNA collected from the brain areas of interest relative to a geometric mean of mRNAs encoding the ribosomal protein 18S. Total mRNA was isolated from the dissected brain areas using the RNeasy Mini Kit (Qiagen, Germany) according to manufacturer's protocol. The purity and quantity of RNA were measured in a nanodrop 1000 (Thermo Scientific, USA). cDNA was synthesized by reverse transcription using 150 ng of RNA in a reaction volume of 20 μl using the High Capacity cDNA Reverse Transcription Kit (Applied Biosystems, USA) according to the manufacturer's protocol. cDNA samples were diluted 1:3 in $H_2O$ before used, and to measure the levels of 18S, a further 2:800 dilution was made. Real-time RT-PCR was conducted using the 7500 Real-Time PCR system (Applied Biosystems) with SYBR Green fluorescent probes using the following conditions: 40 cycles of 95˚C for 15 seconds, 60˚C for 1 minute, and 72˚C for 30 seconds. Forward and reverse primers were ordered from Sigma-Aldrich and used at a concentration of 100 nM each. The following primer pairs were used: gfra1 (Fw: 5′-GAAGATTGCCTGAAGTTTCTGAAT-3′; Rv: 5′-GGTCACATCCGAGCCATT-3′); gdnf (Fw: 5′-GTGACTCCAATATGCCTGAAGA-3′; Rv: 5′-CGCTTGTTTATCTGGTGACCT-3′); and 18S (Fw: 5′- CAATTATTCCCCATGAACG-3′; Rv: 5′- GGCCTCACTAAACCATCCAA-3′). Five biological replicates were performed for each study group. All the experiments were carried out in triplicates for each data point. Data analysis was performed using the 2−ΔCt method, and relative expression levels were calculated for each sample after normalization against the housekeeping gene 18S.

**Protein collection and western blot.** The whole extract and the cytosolic and synaptosome protein fractions were collected using the Syn-PER Synaptic Protein Extraction Reagent (Thermo Fisher, USA) following manufacturer's instructions. The mHb and the IPN of 2 mice per genotype were pulled together due to the small size of the nuclei of interest. Protein concentration was measured with the Pierce BCA Protein Assay Kit (Thermo Fisher, USA), and samples were prepared for SDS-PAGE (20 μg per sample) in SDS sample buffer (Life Technologies) and boiled at 95˚C for 10 minutes before electrophoresis on 4% to 12% pre-cast gels (Sigma-Aldrich) following manufacturer's instructions. Proteins were transferred to polyvinylidene fluoride (PVDF) membranes (Amersham, Germany). Membranes were blocked with 5% nonfat milk in TBST and incubated with primary antibodies overnight in 1% milk. The following primary antibodies were used at the indicated dilutions: Goat anti GFRα1 (1:500, AF560, RnD, USA): goat anti c-RET (1:500, AF482, RnD); rabbit anti NCAM (1:1,000, AB5032, Millipore, USA); Mouse anti PSD95 (1:1,000, MA1-046, Thermo Scientific); rabbit anti GluA1 (1:1,000, AB1504, Millipore); rabbit anti GluA2 (1:1,000, AB1768-I, Millipore); rabbit anti GluA3 (1:1,000, AGC-010, Alomone Labs, Israel); rabbit anti GluA4 (1:1,000, AB1508, Millipore); rabbit anti phospho-Ser831 GluA1 (1:500, 75574, Cell Signaling, USA); rabbit anti phospho-Ser845 GluA1 (1:500, 8084, Cell Signaling); and mouse anti tubulin alpha (1:10,000, T6199, Sigma-Aldrich). Immunoreactivity was visualized using appropriate horseradish peroxidase (HRP)-conjugated secondary antibodies (1:5,000, Dako, Denmark). Immunoblots were developed using the ECL Advance Western Blotting Detection kit (Life Technologies) or the SuperSignal West Femto Maximum sensitivity substrate (Thermo Fisher), and images were acquired with the Image-Quant LAS4000 (GE Healthcare, Japan). Image analysis and quantification of band intensities were done with ImageQuant software (GE Healthcare).

**ELISA for GDNF.**   Different brain areas of 3 C57BL6/J adult mice were dissected, and the whole protein extract was collected by disrupting the tissue in RIPA buffer (Thermo Fisher) with protease inhibitors (Roche, Germany), centrifuging the samples at maximum speed for 10 minutes and collecting the supernatant. Protein concentration was measured using the Pierce BCA Protein Assay Kit (Thermo Fisher). The concentration of GDNF in the samples was measured using the GDNF Mouse Elisa Kit (ab171178, Abcam, United Kingdom) following manufacturer's instructions. All samples were tested in duplicate.

## Histological techniques

**Tissue processing.**   Mice were deeply anaesthetized with isoflurane (Baxter Medical, Sweden) and transcardially perfused with 25 ml of 0.125 M phosphate buffered saline (PBS, pH 7.4, Gibco, United Kingdom) and 40 ml of 4% paraformaldehyde (PFA, Histolab Products, Sweden). Brains were removed, post fixed in PFA overnight and cryoprotected in 30% sucrose in PBS. To obtain 20 μm floating sections, brains were embedded in OCT compound (Sakura, United Kingdom), frozen at −80°C and cut with a cryostat (CryStarNX70, Thermo Scientific). To obtain 30-μm sections, brains were cut in a microtome (Leica SM2000 R, Thermo Scientific, Germany). All sections were stored at −20°C in a cryoprotective solution containing 1% DMSO (Sigma-Aldrich) and 20% glycerol (Sigma-Aldrich) in 0.05 M Tris-HCl pH 7.4 (Sigma-Aldrich).

**General immunofluorescence.**   30-μm brain sections were washed in PBS and blocked for 1 hour in 5% normal donkey serum (NDS, Jackson Immunoresearch) and 0.3% Triton X-100 (Sigma-Aldrich) in PBS. Incubation with the primary antibodies was done overnight at 4°C in blocking solution. After 3 × 10 minutes washes with PBS, the sections were incubated with the corresponding secondary donkey Alexa fluorescent antibodies (1:1,000, Thermo Fisher) and 0.1 mg/ml of 40-6-diamidino-2-phenylindole (DAPI; Sigma-Aldrich) for 2 hours at room temperature (RT). Sections were finally washed with PBS 3 × 10 minutes, mounted on glass slides in a 0.2% solution of gelatin (Sigma-Aldrich) in 0.05 M Tris-HCl buffer (pH 7.4), dried and cover slipped in Dako fluorescent mounting medium. The primary antibodies used in this study were the following: Goat anti GFRα1 (1:500, AF560, RnD); Goat anti ChAT (1:500, AP144P, Millipore); Chicken anti GFP (1:500, ab13970, Abcam); rabbit anti NCAM (1:1,000, AB5032, Millipore); rabbit anti cFOS (1:500, sc-52, Santa Cruz, USA); rabbit anti calretinin (1:500, AB5054, Chemicon, USA); mouse anti calretinin (1:500, 6B3, Swant, Switzerland); rabbit anti GluA1 (1:500, AB1504, Millipore); rabbit anti GluA2 (1:500, AB1768-I, Millipore); rabbit anti GluA3 (1:500, AGC-010, Alomone Labs); rabbit anti GluA4 (1:500, AB1508, Millipore); and rat anti SP (1:500, Millipore, MAB356).

**Retrograde tracing.**   A total of 300 nL of fluorogold (Fluorochrome) were injected unilaterally in the mHb of *Gfra1*^(GFP) and *Ret*^(GFP) adult mice at the following coordinates obtained from the Paxinos and Franklin atlas: 10° angle, AP −1.94 mm, ML +0.75 mm, and DV −2.3 mm. Mice were perfused 2 weeks later.

**Immunofluorescence for synaptic markers.**   20-μm brain sections were washed in PBS, blocked for 1 hour at RT in 20% NDS in PBS, incubated with the primary antibodies at 4°C overnight in a solution containing 10% NDS and 0.3% TX-100, washed 3 x 1 hour in PBS, incubated in the same solution as the primary antibodies with the corresponding secondary donkey Alexa fluorescent antibodies (1:1,000, Thermo Fisher) for 2 hours, washed 3 x 1 hour in PBS, counterstained with DAPI for 10 minutes, and washed for 2x30 minutes in PBS. Sections were finally mounted on glass slides in a 0.2% solution of gelatin (Sigma-Aldrich) in 0.05 M Tris-HCl buffer (pH 7.4), dried, and cover slipped in Dako fluorescent mounting medium. The primary antibodies used were guinea pig anti VGlut1 (1:500, AB2251, Millipore); Mouse anti VGlut2 (1:500, 135421, SYSY); rabbit anti PSD93 (1:500, 124102, SYSY); Goat anti GFRα1

(1:500, AF560, RnD); guinea pig anti VGAT (1:500, 131004, SYSY); Mouse anti gephyrin (1:500, 147111, SYSY); and goat anti synapsin Ia/b (1:500, sc8295, Santa Cruz). PSD93 antibody was used to label the postsynaptic site of the glutamatergic synapses since we found that PSD95 antibody did not work for immunohistochemistry.

**PLA.** PLA was performed according to manufacturer's instructions (Duolink; Sigma-Aldrich). Briefly, 30-μm coronal sections containing the mHb and the IPN were blocked and permeabilized in 5% NDS and 0.3% TX-100 in PBS for 1 hour and incubated with rabbit anti GluA1 (1:400, AB1504, Millipore) and mouse anti GluA2 (1:400, Abcam, ab192760) or rabbit anti GluA1 (1:400, AB1504, Millipore) and goat anti GluA4 (1:400, ab115322, Abcam) overnight at 4˚C. Mouse anti NMDAR2B (1:400, 610416, BD) and mouse anti DARP32 (1:400, sc-271111, Santa Cruz) were used as negative controls. Then sections were washed with PBS, incubated with minus and plus PLA probes for 1 hour at 37˚C, washed with buffer A, incubated with Ligation-Ligase solution for 30 minutes at 37˚C, washed again with buffer A, and finally incubated with the Amplification-Polymerase solution (Duolink In Situ detection reagents Orange) for 100 minutes at 37˚C. After washing with buffer B, sections were incubated for 10 minutes with DAPI (1:10,000 in buffer B), mounted on glass slides, and coverslipped with Dako mounting media.

**In situ hybridization.** In situ hybridization was performed in 30-μm coronal sections containing the mHb and the IPN from C57BL6/J 3-month-old mice using RNAscope technology (Advanced Cell Diagnostics Biotechne, USA), following the manufacturer's protocol. The following probe was used: GFRa1 (Mm-Gfra1-C2, Cat No. 431781-C2).

**Image acquisition and analysis.** All fluorescent images were captured with a Carl Zeiss LSM 710 confocal microscope using ZEN2009 software (Carl Zeiss, Germany). All images from the same brain area and experiment were acquired using the same parameters. For imaging of the mHb and the IPN, 1 section every 120 μm was used for all experiments (6 to 8 sections per mHb and 4 to 5 sections per IPN). For the synaptic markers and the PLA experiments, 4 mHb images were obtained with the 63× objective per section (1 dorsal and 1 ventral per mHb). For the IPN, 6 images were obtained per section (2 dorsal, 2 ventral, and 2 lateral). Image analysis was made with ImageJ and Fiji softwares. Quantification of the number of synaptic puncta and PLA dots was done in imageJ with the plugin synaptic puncta analyzer [54]. In brief, background was subtracted from the image and the same intensity threshold and range of particle size was used to analyze all images from the same experiment.

**Transmission electron microscopy.** $Gfra1^{+/+}$, $Gfra1^{CreERT2/+}$, and $Gfra1^{Cre/LoxP}$ 3-month-old mice were treated with tamoxifen and perfused 1 month later with 2.5% glutaraldehyde and 1% PFA in 0.1 M PB. Brains were removed and left on the same fixative overnight at 4˚C. The day after 300 μm slices were obtained in a vibratome (Leica VT1000S), and the area corresponding to the mHb and the IPN was cut out. Subsequent tissue preparation and image acquisition was done at the electron microscopy unit (EMil) of Karolinska Institutet. The tissue blocks were washed in 0.1 M PB; postfixed in 2% OsO4 in 0.1 M PB for 2 hours at 4˚C and then incubated with (i) 70% EtOH for 30 minutes at 4˚C; (ii) 95% EtOH for 30 minutes at 4˚C; (iii) 100% EtOH with 0.5% uranil acetate for 20 minutes at RT; (iv) acetone 2 × 15 minutes RT; (v) LX-112/acetone (1:2) for 4 hours at RT; (vi) LX-112/acetone (1:1) overnight at RT; (vii) LX-112/acetone (1:2) for 8 hours at RT; and (viii) LX-112 overnight at RT and finally embedded in LX-112 resin at 60˚C. Ultrathin sections were cut with a EM UC6 (Leica), and images were acquired using both the FEI Tecnai 12 Spirit BioTwin transmission electron microscope (FEI, Eindhoven, the Netherlands) and the Hitachi HT7700 (Hitachi High-Technologies, Japan) transmission electron microscope both operated at 100 kV and equipped with 2kx2k Veleta CCD cameras (Olympus Soft Imaging Solutions GmbH, Germany).

Analysis of TEM images was performed using ImageJ software. For the mHb, 25 images of each subnuclei (dmHb and vmHb) were analyzed. For the IPN, 15 to 25 images of each subnuclei (dorsal and ventral IPN) were analyzed. Synapses, which had clear synaptic vesicles and PSD, were visually identified and counted in all images. The PSD length, the number of synaptic vesicles per vesicle pool, and the wide of the synaptic cleft were quantified manually in all synapses using ImageJ software. For calculation of the wide of the synaptic cleft, the area of the synapse was divided by the PSD length. The average of each parameter was calculated for each animal, and the results were expressed as mean ± SEM of 3 animals per genotype. In the mHb, a total of 240, 207, and 115 synapses were analyzed in WT, Het, and KO mice, respectively. In the IPN, a total of 216, 157, and 104 S-synapses were analyzed in WT, Het, and KO mice, respectively. Crest synapses were excluded from the structural analysis but included in the counts of the number of synapses. Crest synapses were identified in a small percentage of images of the ventral IPN, showing a characteristic morphology in which 2 mHb axons synapse on opposite sides of a dendritic process, containing a PSD and vesicle pool on opposing sides of the dendritic process [34].

## Chemogenetics and optogenetics

**Chemogenetics.**  *Gfra1*$^{+/+}$, *Gfra1*$^{CreERT2/+}$, and *gfra1*$^{CreERT2/LoxP}$ adult mice were stereotaxically injected: (i) in the posterior septum with 500 nL of a 1:1 mixture of AAV2-CMV-Cre-GFP ($3.7 \times 10^{12}$ genomic copies per mL; UNC Gene therapy vector core) and AAV2-hSyn-DIO-hM3D(Gq)-mCherry ($6.1 \times 10^{12}$ genomic copies per mL; UNC Gene therapy vector core) or the AAV2-CMV-Cre-GFP virus alone in the following coordinates obtained from the Paxinos and Franklin atlas: 10˚ angle; AP -0.1 mm, ML +0.5 mm, DV– 2.58 mm; and (ii) bilaterally in the mHb with 300 nL of AAV2-hSyn-DIO-hM3D(Gq)-mCherry or 300nL of a control AAV2-CMV-GFP virus ($5.4 \times 10^{12}$ genomic copies per mL; UNC Gene therapy vector core) at the following coordinates: 10˚ angle, AP −1.94 mm, ML ±0.75 mm, DV −2.3 mm). One week after, viral delivery animals were treated with tamoxifen to induce global ablation of *Gfra1* expression. Activation of hM3Dq was induced by an i.p. injection of CNO (Tocris, United Kingdom) dissolve in 2% DMSO in NaCl 0.9% serum (Braun, Germany) 28 days after the first dose of tamoxifen, and the animals were perfused 1 hour later.

**Optogenetics.**  Ex vivo electrophysiological experiments were performed as described previously [12]. Briefly, adult *gfra1*$^{+/+}$, *Gfra1*$^{CreERT2/+}$, and *gfra1*$^{CreERT2/LoxP}$ mice (3 to 4 months old) were injected with 200/300nL of a 1:1 mix of AAV2/1.EF1a.DIO.hChR2.eYFP and AAV2/9.CMV.Cre (with final titers of 2.02e13 and 4.90e12 GC/mL, respectively; Vector Core of the University of Pennsylvania, USA) in the posterior septum at the following coordinates, from bregma: ML 0.0 mm; AP 0.13 to 0.2 mm; and DV −2.6 mm. Mice were treated with tamoxifen (4 consecutive daily i.p. injections, 100 mg/kg) starting 1 week after surgery. Minimum 4 weeks following the viral injections, slices were obtained and whole-cell, voltage-clamp recordings of mHb neurons were performed at 32˚C to 34˚C in a bicarbonate-buffered saline containing (in mM): 116 NaCl, 2.5 KCl, 1.25 NaH$_2$PO$_4$, 26 NaHCO$_3$, 30 glucose, 1.6 CaCl$_2$, 1.5 MgCl$_2$, and $5 \times 10^{-5}$ minocycline (bubbled with 95% O$_2$, 5% CO$_2$), and using an intracellular solution containing (in mM): 135 CsMeSO$_3$, 10 TEA-Cl, 4.6 MgCl$_2$, 10 HEPES, 10 K$_2$-creatine phosphate, 0.5 EGTA, 4 Na$_2$-ATP, 0.4 Na$_2$-GTP, 0.1 spermine and 1 QX-314, pH 7.35 and mOsm approximately 300. EPSCs were elicited by brief (2 to 4 ms long) pulses of blue light provided by a 470-nm wavelength diode (Thorlabs, France) coupled to the slice chamber via the epifluorescence pathway of the microscope. EPSCs were recorded in the constant presence in the bath of the GABAa and glycine receptor blockers SR-95501 and strychnine respectively. Drugs were bath applied: AMPAR antagonist NBQX (10 µM), NMDA antagonist D-APV

(50 μM), SR-95501 (5 μM) from Tocris Bioscience, and strychnine (2 μM) from Sigma-Aldrich. Acquired with an EPC-10 double amplifier using the PatchMaster software (both from Heka Elektronik, Germany), the recordings were sampled at 10/40 KHz, filtered at 10 KHz and analyzed offline with custom routines written in Igor (Wavemetrics, USA). NMDAR-mediated EPSCs were quantified at +50 mV, as the response amplitudes at the 7 ms-post peak time point, at full decay of the AMPAR component. The amplitudes of the AMPAR-mediated EPSC components at the holding potentials of +50 mV and −60 mV were measured at the peaks of the responses following bath application of the NMDA antagonist APV. Their ratio is the R.I. of the AMPAR currents before D-APV application.

## Behavioral studies

All behavioral experiments were performed between 9 AM and 2 PM, under low light conditions and blind to the genotype and treatment. The only sex differences found were on body weight; therefore, the results of both sexes were pooled together. In all cases, mice were acclimatized to the behavioral room for at least 20 minutes before testing and all apparatus were cleaned with 70% ethanol between animals. Behavioral evaluation started 2 weeks after the first tamoxifen injection or 2 weeks after AAV2.Cre virus delivery into the mHb. The different behavioral tests were performed in the following order with 4 days between each test: open field test, elevated plus maze, innate fear response, and passive avoidance. In a different set of mice, the following behavioral tests were performed with 7 days between each test: open field test, context fear conditioning, and cued fear conditioning.

**Open field test.** Mice were placed in the center of a 48 cm × 48 cm transparent acrylic box equipped with lightbeam strips (TSE Actimot system) and allowed to move freely for 10 minutes. General locomotor parameters including distance traveled, time moving and distance, and time spent in the center of the arena were calculated using Actimot Software.

**Elevated plus maze.** The apparatus consisted of a cross-shaped maze with 2 open arms opposite to 2 arms enclosed by lateral walls (70 cm × 6 cm × 40 cm) elevated 70 cm above the floor. Each animal was placed in the central square (6 cm × 6 cm) facing one of the open arms away from the experimenter and allowed to explore the maze for 5 minutes. The movement of the animals was recorded using the Ethovision XT-10 tracking software and the number of entries, the staying time in each of the arms and the distance covered were analyzed.

**Innate fear measurement.** The innate freezing response of the animals after a foot shock was evaluated using a TSE multiconditioning system (TSE Systems, Germany). Each animal was placed in a squared behavioral arena made of plexiglass with 4 transparent walls, and after 3 minutes of habituation, the mouse received 3 consecutive foot shocks separated by 1 minute intervals (0.7 mA; 1 second). Animals were left in the arena for 2 extra minutes, and the percentage of freezing behavior during each minute interval was analyzed.

**Passive avoidance.** Passive avoidance is a fear-motivated learning task in which the animal suppresses a motor response in order to avoid exposure to a dark compartment that has been previously coupled with a foot shock. The apparatus chamber used in this test was composed by 2 compartments (a preferred dark chamber and an illuminated one with 1000 Lux) connected through an automatic door (TSE multiconditioning system, TSE Systems). On the acquisition/conditioning day, animals were place on the illuminated compartment and 30 seconds after the door was opened. When the animals crossed to the preferred dark chamber, the door was closed, and they received a mild foot shock (1 second, 0.7 mA). The day after the mice were placed again on the illuminated compartment (1000 Lux), 30 seconds later, the door was opened, and the latency to enter the dark side was measured with a maximum latency time of 600 seconds.

**Fear conditioning.** In the Fear Conditioning paradigm, mice associate a neutral conditional stimulus (tone or context) with an aversive unconditional stimulus (a mild electrical foot shock). In the memory test, the animal shows a conditional response (freezing response) to the cue or the context. The TSE multiconditioning system apparatus from TSE systems was used for both the context and the cued fear conditioning.

In the context fear conditioning, mice were exposed to context A (square box with 1 transparent side and 3 black sides) for 3 minutes ending with a foot shock (2 seconds, 0.7 mA). The day after, the animals were exposed to the same context A for 3 minutes, and the freezing response was measured using the automatic counting system in the TSE system.

In the cued fear conditioning, mice were exposed to context A (square box with 1 transparent side and 3 black sides) for 2 minutes, and then a tone was played for 20 seconds (80 Db, 3,000 Hz) ending with a foot shock (0.7 mA, 2 seconds). The day after the animals were placed in context B (a transparent cylinder box) and after 1 minute, they were exposed to the same tone for 5 minutes. The freezing response was measured using the automatic counting system in the TSE system.

## Statistical analysis

Data are expressed as mean and S.E.M. No statistical methods were used to predetermine sample sizes, but our sample sizes are similar to those generally used in the field. Following normality test and homogeneity variance (F-test or Kolmogorov–Smirnov test with Dallal–Wilkinson–Lillie for *p*-value), group comparison was made using an unpaired Student *t* test, 1-way or 2-way ANOVA as appropriate followed by Bonferroni or Tukey multiple comparison test for normally distributed data. Mann–Whitney U test or Kruskal–Wallis followed by Dunn multiple comparison test was used on non-normal distributed data. Differences were considered significant for $p < 0.05$. The raw data used to generate all graphs in this study are available at https://figshare.com/projects/Raw_data_Fernandez-Suarez_et_al_2021/123406.

## Supporting information

**S1 Fig. Characterization of GFRα1 and co-receptor expression in the adult mouse brain.**
**(A)** dTomato epifluorescence (red) in coronal sections of *Gfra1^dTOM^* mouse brain injected with tamoxifen at 3 months counterstained with DAPI (blue). Scale bars, 200μm. **(B, C)** Expression levels of *gfra1* (B) and *gdnf* (D) mRNAs quantified by real-time RT-PCR and normalized against *18S* in brain areas dissected from 3-month-old C57BL6/J mice (*n* = 5). **(D)** Immunoblot of whole protein extracts of 3-month-old C57BL6/J mice probed for GFRα1. β-actin was probed as loading control. **(E)** Concentration of GDNF in whole protein extracts of 3-month-old C57BL6/J mice quantified by ELISA (*N* = 3 mice). **(F, G)** dTomato epifluorescence (red), GFP (green), and NCAM (green) immunolabeling in coronal sections of the mHb and FR (F) and IPN (G) of *gfra1^dTOM^Ret^GFP^* mouse injected with tamoxifen at 3 months. Scale bars, 200 μm (mHb) and 30 μm (inset 1), 50 μm (FR), 200 μm (IPN), and 40 μm (insets 2 and 3). The data underlying this figure can be found at https://figshare.com/projects/Raw_Data_Fernandez-Suarez_et_al_2021/123406. Cx, cortex; FR, fasciculus retroflexus; GDNF, glial cell line–derived neurotrophic factor; GFRα1, glial cell–derived neurotrophic factor receptor alpha 1; Hb, habenula; Hipp, hippocampus; IPN, interpeduncular nucleus; LHb, lateral habenula; LS, lateral septum; mHb, medial habenula; MS, medial septum; NCAM, neural cell adhesion molecule; OB, olfactory bulb; PS, posterior septum; SNpc, substantia nigra pars compacta; SNr, substantia nigra reticulata; STR, striatum.
(TIF)

**S2 Fig. Characterization of *Gfra1* expression by RNAscope in the mHb and the IPN of adult C57BL6/J mice.** Representative images of the dorsal and vmHb and IPN showing RNA-scope for *Gfra1* (red), immunolabeling for ChAT (green), and counterstaining with DAPI in brain sections from a C57BL6/J adult mouse. Scale bars, 20 μm. ChAT, choline acetyltransferase; IPN, interpeduncular nucleus; mHb, medial habenula; vmHb, ventral medial habenula. (TIF)

**S3 Fig. Expression of GFRα1 and co-receptors in the TS and BAC. (A, E)** Schematic representation of a sagittal brain section of a 3-month-old *Gfra1*$^{GFP}$ (A) or *Ret*$^{GFP}$ (E) mouse injected with FG in the mHb. **(B, F)** GFP (green) immunolabeling in coronal sections of the mHb of a *Gfra1*$^{GFP}$ (B) or a *Ret*$^{GFP}$ (F) mouse injected with FG (blue) in the mHb. Scale bars, 100 μm. **(C, D, G, H)** CR (red) and GFP (green) immunolabeling in coronal sections of the TS and BAC from a *Gfra1*$^{GFP}$ (C, D) or a *Ret*$^{GFP}$ (G, H) mouse injected with FG (blue) in the mHb. Scale bars, 200 μm (C, G) and 50 μm (D, H). **(I)** dTomato epifluorescence (red) and GFP (green) immunolabeling in a coronal section counterstained with DAPI (blue) of the TS of a *Gfra1*$^{dTOM}$*Ret*$^{GFP}$ mouse treated with tamoxifen at 3 months. Scale bars, 200 μm and 50 μm (inset). **(J, K)** NCAM (red) and CR (green) immunolabeling in coronal sections of the TS and BAC of a 3-month-old C57BL6/J mouse. Scale bars, 100 μm (J) and 20 μm (K). BAC, bed nucleus of the anterior commissure; CR, calretinin; FG, fluorogold; GFRα1, glial cell–derived neurotrophic factor receptor alpha 1; mHb, medial habenula; NCAM, neural cell adhesion molecule; TS, triangular septum. (TIF)

**S4 Fig. Global and mHb-specific induction of *gfra1* deletion in adult mice. (A)** Schematics of the global *gfra1* KO mouse model. **(B)** dTomato epifluorescence (red) in coronal sections of the mHb and the IPN of *gfra1*$^{dTOM}$ mouse treated with tamoxifen or vehicle at 3 months. Images corresponding to the animal injected with vehicle show also counterstaining with DAPI (blue). Scale bars, 200 μm. **(C)** Immunolabeling for GFRα1 (red) in coronal sections of the mHb and the IPN of WT, Het, and KO mice. Scale bars, 100 μm. **(D, E)** Quantification (± SEM) of GFRα1 OD in the mHb (D) and the IPN (E) of WT, Het, and KO mice normalized against the WT group. $N$ = 4–6 animals/group; 1-way ANOVA analysis followed by Tukey post hoc test at each time point; *$P < 0.05$; **$P < 0.01$; ***$P < 0.001$. **(F, G)** Immunoblots of total protein fraction from mHb and IPN of WT, Het, and KO mice probed for GFRα1 (F) and αtubulin. Quantifications (± SEM) of GFRα1 levels were corrected for αtubulin levels and normalized to levels in WT samples. $N$ = 6 mice per group; 1-way ANOVA followed by Tukey post hoc test; * $p < 0.05$; **$p < 0.01$; ****$p < 0.0001$. **(H)** Quantification by qPCR of *Gfra1* mRNA expression levels (± SEM) in the mHb and the IPN of WT, Het, and KO mice. *Gfra1* levels were corrected for *18S* levels and normalized to levels in the mHb of WT samples. $N$ = 5 mice per group, 2-way ANOVA followed by Tukey post hoc test; * $P < 0.05$; ***$P < 0.001$; ****$P < 0.0001$. **(I)** GFP (green) immunolabeling in coronal sections counterstained with DAPI (blue) of the mHb of a *R1CG*$^{lx/lx}$ mouse injected with an AAV.CMV.Cre virus in the mHb. Scale bar, 100 μm. **(J)** GFRα1 immunolabeling (red) in coronal sections of the mHb and the IPN of *gfra1*$^{lx/lx}$ mice injected with vehicle (mHb.WT) or AAV.Cre (mHb.KO) in the mHb. Scale bars, 50 μm (mHb) and 200 μm (IPN). **(K)** Quantification (± SEM) of GFRα1 OD in the mHb and the IPN of mHb.WT and mHb.KO mice. $N$ = 5 animals per group; Student *t* test; **$P < 0.01$; ****$P < 0.0001$. The data underlying this figure can be found at https://figshare.com/projects/Raw_Data_Fernandez_Suarez_et_al_2021/123406. GFRα1, glial cell–derived neurotrophic factor receptor alpha 1; HET, heterozygous; IPN, interpeduncular nucleus; KO, knockout; mHb, medial habenula; OD, optical density; WT, wild-type. (TIF)

**S5 Fig. Quantification of VGlut2, VAChT, VGAT, and gephyrin puncta and number of DAPI cells after global ablation of GFRα1. (A)** VGlut2 (red) immunostaining in coronal sections of the mHb and the IPN of WT, Het, and KO mice. Scale bar, 10 μm. **(B)** Quantification (± SEM) of puncta with immunoreactivity for VGlut2 in mHb and IPN. *N* = 5 mice per group (25–30 images per mouse); 1-way ANOVA followed by Tukey post hoc test; * $P < 0.05$, ** $P < 0.01$, *** $P < 0.001$. **(C)** VGAT (red) and gephyrin (green) immunostaining in coronal sections of the mHb of WT, Het, and KO mice. Scale bar, 10 μm. **(D)** Quantification (± SEM) of puncta with immunoreactivity for VGAT and gephyrin in the mHb. *N* = 4–5 mice per group (25–30 images per mouse). **(E)** VAChT (red) immunostaining in coronal sections of the IPN of WT, Het, and KO mice. Scale bar, 20 μm. **(F)** Quantification (± SEM) of puncta with immunoreactivity for VAChT in the dorsal, ventral and lateral IPN. *N* = 7, 6, and 5 mice in WT, Het, and KO mice, respectively (25–30 images per mouse per structure). **(G)** DAPI (blue) counterstaining in coronal sections of the mHb of WT, Het, and KO mice. Scale bar, 20 μm. **(H)** Quantification (± SEM) of number of DAPI cells in the mHb. *N* = 4 mice per group (32 images per mouse). The data underlying this figure can be found at https://figshare.com/projects/Raw_Data_Fernandez-Suarez_et_al_2021/123406. GFRα1, glial cell–derived neurotrophic factor receptor alpha 1; HET, heterozygous; IPN, interpeduncular nucleus; KO, knockout; mHb, medial habenula; VGlut2, vesicular glutamate transporter 2; WT, wild-type. (TIF)

**S6 Fig. Control groups for the TS → mHb chemogenetic experiment.** Mice were injected with viral vectors in the TS and treated with tamoxifen and CNO as indicated in each panel. **(A–D)** Viral transfection efficiency in the TS of WT, Het, and KO mice. mCherry epifluorescence (red), GFP (gray), and immunolabeling for cFOS (green) in coronal sections containing the TS from WT, Het, and KO mice injected with a combination of the Cre and DREADD viruses into the TS (A; scale bars, 200 μm). Graphs show the quantification (± SEM) of the number of GFP⁺ cells (B) and cFOS⁺ cells (C) in the TS and the OD for mCherry in the terminals in the mHb (D) of WT, Het, and KO mice. *N* = 6 mice per group (4–6 TS sections and 6–8 mHb sections per mouse). **(E–H)** cFos expression after CNO treatment in the absence of the DREADD. GFP (gray), immunolabeling for cFOS (green) and counterstaining with DAPI (blue) in coronal sections containing the TS (E) or the mHb (F) from a Het mouse injected with a AAV.Cre.GFP virus into the TS (scale bar TS = 200 μm; mHb = 100 μm). Graphs show the quantification (± SEM) of the number of cFOS⁺ cells in the TS (G) and in the mHb (H) after CNO treatment (5 mg/kg; i.p.). *N* = 3 mice per group (4–6 TS sections TS and 6–8 mHb sections per mouse). The data underlying this figure can be found at https://figshare.com/projects/Raw_Data_Fernandez-Suarez_et_al_2021/123406. CNO, clozapine N-oxide; DREADD, designer receptor exclusively activated by designer drug; HET, heterozygous; KO, knockout; mHb, medial habenula; OD, optical density; TS, triangular septum; WT, wild-type. (TIF)

**S7 Fig. Control groups for the mHb → IPN chemogenetic experiment.** Mice were injected with viral vectors in the mHb and treated with tamoxifen, CNO, or vehicles as indicated in each panel. **(A–C)** Viral transfection efficiency in the mHb of WT, Het, and KO mice. mCherry epifluorescence (red), immunolabeling for c-Fos (green), and counterstaining with DAPI (blue) in coronal sections containing the mHb from WT, Het, and KO mice injected with the DREADD virus into the mHb (A; scale bars, 100 μm). Graphs show the quantification (± SEM) of the OD for mCherry (B, *N* = 6 mice per group, 10–12 sections per mouse) and the number of cFOS⁺ cells (C, *N* = 7, 11, 12) in the mHb of WT, Het, and KO mice. **(D–F)** cFOS expression in the mHb and IPN after CNO treatment in the absence of the DREADD.

Immunolabeling for cFOS (red) and counterstaining with DAPI (blue) in the mHb and IPN (D) of a mouse injected with an AAV2.GFP virus in the mHb and treated with CNO. Graphs show the quantification (± SEM) of cFOS$^+$ cells in the mHb (E) and the IPN (F). Scale bars, 70 μm (mHb) and 100 μm (IPN). **(G)** mCherry epifluorescence (red), cFOS immunostaining (green), and counterstaining with DAPI in the mHb of a mouse injected with the DREADD virus and treated with the vehicle of tamoxifen showing no expression of the DREADD neither cFOS after CNO treatment. Scale bar, 100 μm. **(H)** mCherry epifluorescence (red) and cFos immunostaining (green) in the mHb and the IPN of a mouse injected with the DREADD virus showing no cFOS expression in the absence of CNO. Scale bars, 100 μm (mHb) and 200 μm (TS). The data underlying this figure can be found at https://figshare.com/projects/Raw_Data_Fernandez-Suarez_et_al_2021/123406. CNO, clozapine N-oxide; DREADD, designer receptor exclusively activated by designer drug; HET, heterozygous; IPN, interpeduncular nucleus; KO, knockout; mHb, medial habenula; OD, optical density; TS, triangular septum; WT, wild-type. (TIF)

**S8 Fig. Characterization of GluA1-4 expression in the mHb and IPN of C57BL6/J adult mice. (A)** GluA1-4 immunostaining (red) and counterstaining in DAPI (blue) in coronal sections of the mHb and the IPN of a 3-month-old C57BL6/J mouse. Scale bars, 150 μm (mHb) and 300 μm (IPN). **(B)** Immunoblots of WE, Cyt, and Syn protein extracts from the mHb and IPN of 3-month-old C57BL6/J mice probed against GluA1-4. PSD95 and tubulin were probed as loading controls. **(C)** PLA signals (red) for GluA1-GluA2 and GluA1-GluA4 complexes in coronal sections of mHb and IPN from 3-month-old C57BL6/J mice. Counterstaining with DAPI appears in blue. NMDAR2B and DARP32 were used as negative controls. Scale bars, 20 μm. **(D, E)** Quantification (± SEM) of PLA puncta for GluA1-GluA2 and GluA1-GluA4 complexes in the dorsal and vmHb (D) and the lateral and core IPN (E) from 3-month-old C57BL6/J mice. $N$ = 4 mice (16 images per mHb and 18 images per IPN). Student $t$ test; $^*$ $P$ < 0.05; $^{**}$ $P$ < 0.01. The data underlying this figure can be found at https://figshare.com/projects/Raw_Data_Fernandez-Suarez_et_al_2021/123406. Cyt, cytosolic protein fraction; IPN, interpeduncular nucleus; mHb, medial habenula; PLA, proximity ligation assay; Syn, synaptosome protein fraction; vmHb, ventral medial habenula; WE, whole protein extract. (TIF)

**S9 Fig. Acute loss of gfra1 in adult mHb neurons does not alter the behavioral responses in the open field, the innate freezing response, or a cued fear conditioning paradigm.** (A, B) Total distance, time moving and distance, and time spent in the center of the open field arena (± SEM) for WT, Het, and KO mice (A) and mHb.WT and mHb.KO mice (B). $N$ = 28–31 mice per group, Kruskal–Wallis test followed by Dunn multiple comparison test (A). $N$ = 12 mice per group, Student $t$ test (B). (C, D) Distance covered in the open and closed arms of the EPM for WT, Het, and KO mice ($N$ = 10–12 animals per group) (C) and mHb.WT and mHb.KO mice ($N$ = 11–12 animals per group) (D). (E, F) Correlation analysis between the distance covered in the open field test (OF) and in the EPM for WT, Het, and KO mice (E) and mHb.WT and mHb.KO mice (F). (G) Freezing responses (± SEM) after 3 consecutive foot shocks for WT, Het, and KO mice ($N$ = 10–13 mice per group) and mHb.WT and mHb.KO mice ($N$ = 12 mice per group). Two-way ANOVA followed by Tukey post hoc test. (H) Percentage of freezing in the training day of the context fear conditioning test for WT, Het, and KO mice ($N$ = 8–10 mice per group, 1-way ANOVA) and mHb.WT and mHb.KO mice ($N$ = 9–10 mice per group, Student $t$ test). (I, J, K) Schematic (I), freezing timecourse and total freezing time (± SEM) for WT, Het, and KO mice (J) and mHb.WT and mHb.KO mice (K) in the cued fear conditioning test. $N$ = 8–10 mice per group, 2-way ANOVA and 1-way ANOVA (J). $N$ = 9–10 mice per group, 2-way ANOVA and Student $t$ test (K). The data underlying this figure can be

found at https://figshare.com/projects/Raw_Data_Fernandez-Suarez_et_al_2021/123406.
EPM, elevated plus maze; HET, heterozygous; KO, knockout; mHb, medial habenula; WT,
wild-type.
(TIF)

**S1 Table. Statistics, Figs 1–6.**
(PDF)

**S2 Table. Two-way ANOVA analysis, Figs 1–6.**
(PDF)

**S3 Table. Statistics, S1–S9 Figs.**
(PDF)

**S4 Table. Two-way ANOVA analysis, S1–S9 Figs.**
(PDF)

**S1 Raw Images. Original images from the western blots showed in the study.**
(PDF)

## Acknowledgments

The authors would like to thank Maria Christina Sergaki and Wei Wang for technical assis-
tance during the early stages of the project; Mart Saarma and Jaan-Olle Andressoo (University
of Helsinki, Finland) for *Gfra1*fx/fx mice; Minmin Luo for his comments on the manuscript;
Linda Thors, Emma Wallet, and personnel from the KMB animal facility of Karolinska Institu-
tet for their help with animal care; IIna Eleonoora Korkala for DAPI cell counts; and all CIB
Lab members for comments and suggestions.

## Author Contributions

**Conceptualization:** Diana Fernández-Suárez, Carlos F. Ibáñez.

**Formal analysis:** Diana Fernández-Suárez.

**Funding acquisition:** Carlos F. Ibáñez.

**Investigation:** Diana Fernández-Suárez, Favio A. Krapacher, Katarzyna Pietrajtis, Annika
Andersson, Lilian Kisiswa, Alvaro Carrier-Ruiz, Marco A. Diana.

**Methodology:** Diana Fernández-Suárez, Marco A. Diana, Carlos F. Ibáñez.

**Project administration:** Diana Fernández-Suárez, Carlos F. Ibáñez.

**Resources:** Carlos F. Ibáñez.

**Supervision:** Diana Fernández-Suárez, Carlos F. Ibáñez.

**Validation:** Diana Fernández-Suárez.

**Visualization:** Diana Fernández-Suárez, Carlos F. Ibáñez.

**Writing – original draft:** Diana Fernández-Suárez, Carlos F. Ibáñez.

**Writing – review & editing:** Diana Fernández-Suárez, Favio A. Krapacher, Katarzyna Pietraj-
tis, Annika Andersson, Lilian Kisiswa, Alvaro Carrier-Ruiz, Marco A. Diana, Carlos F.
Ibáñez.

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
