## [Editor Report · Decision Letter 0]

29 Jun 2021

Dear Dr Fernandez-Suarez, 

Thank you for submitting your manuscript entitled "Adult medial habenula neurons require GDNF receptor GFRalpha1 for synaptic stability and function" for consideration as a Research Article by PLOS Biology.

Your manuscript has now been evaluated by the PLOS Biology editorial staff as well as by an academic editor with relevant expertise and I am writing to let you know that we would like to send your submission out for external peer review. 

Please re-submit your manuscript within two working days, i.e. by Jul 01 2021 11:59PM.

Given the disruptions resulting from the ongoing COVID-19 pandemic, please expect delays in the editorial process. We apologise in advance for any inconvenience caused and will do our best to minimize impact as far as possible. ALso thank you for your patience as we completed our editorial process, and please accept my apologies for the delay in providing you with our decision. We have had difficulty recruiting an academic editor.

Kind regards,

Lucas Smith

Associate Editor

PLOS Biology

lsmith@plos.org

---

## [Decision Letter · Decision Letter 1]

4 Aug 2021

Dear Dr Fernandez-Suarez,

Thank you for submitting your manuscript "Adult medial habenula neurons require GDNF receptor GFRalpha1 for synaptic stability and function" for consideration as a Research Article at PLOS Biology. Your manuscript has been evaluated by the PLOS Biology editors, an Academic Editor with relevant expertise, and by several independent reviewers.

The reviews of your manuscript are appended below. As you will see from their comments, while the reviewers have feel that the paper is interesting, they have also raised a number of concerns that will need to be addressed before we can consider your manuscript for publication. Reviewer 1 notes that some of the conclusions and interpretations made in the study are not fully justified, have inconsistencies, or are not currently adequately supported by the data. Reviewer 2 has raised the need for additional discussion of developmental phenotypes and validation of the GDNF antibody used here, and reviewer 3 raises a number of additional analyses that would be important to strengthen the study. The reviewers also note missing references, statistical analyses, and methodological description.

In light of the reviews, we will not be able to accept the current version of the manuscript, but we would welcome re-submission of a much-revised version that takes into account the reviewers' comments and addresses their concerns with new experimental data where appropriate. We cannot make any decision about publication until we have seen the revised manuscript and your response to the reviewers' comments. Your revised manuscript is also likely to be sent for further evaluation by the reviewers.

We expect to receive your revised manuscript within 3 months. 

**IMPORTANT - SUBMITTING YOUR REVISION**

*Re-submission Checklist*

*Published Peer Review*

*PLOS Data Policy*

*Blot and Gel Data Policy*

Sincerely,

Lucas Smith

Associate Editor

PLOS Biology

lsmith@plos.org

REVIEWS:

Reviewer #1: This manuscript entitled "Adult medial habenula neurons-----------------------------for synaptic stability and function" by Fernandez-Suarez et al., analyzes the role of GFRa1 in the septal-mHb-IPN pathway. This study demonstrates a robust expression of GFRa1 in the mHb. Authors have used a wide range of genetic, chemogenetic, optogenetic, imaging, electrophysiological and behavioral tools to undertake this study. While this study has addressed the effect of GDNF signaling on some of the important synaptic and behavioral aspects of the septal-mHb-IPN pathway, the incorrect/less rigorous interpretation of some of the results diminish enthusiasm for the current version of this manuscript. Also, the discussion section is unfocused and does not adequately convey the importance of the current findings. 

Major comments

1. Lack of GFRa1 in the IPN neurons: The expression of tdTomato in the IPN of Gfra1tdTom mice might be due to the mHb axons as the authors claim (as suggested by the mRNA results). Authors use GAD67 cre mice to confirm that only a few IPN neurons show Gfra1. However, it is unclear how much of the GAD67cre neurons represent IPN neurons. Multiple subtypes of GABAergic neurons might be present in the IPN. Therefore, a confirmation is needed that the majority if not all of the IPN GABAergic neurons are labeled in GAD67 cre mice. A relevant issue here is the co-localization of PSD93 and Gfra1, yet the authors claim Gfra1 is pre-synaptic in the IPN. Fig 1D shows GFP in areas of the IPN where there is no ChAT or SP staining. It must be noted that the mHb axonal pattern in the IPN (Ren et al., Neuron, 69: 445-452, 2011; Frahm et al., eLife, 4: e11396, 2015) is very distinct from the distribution of GFRa1 in the IPN. The expression of GFRa1 in the lateral part of the IPN is much more intense. These issues need to be addressed when interpreting these results. 

2. Since the authors mention S and crest synapses in the IPN, it is important that figures are labeled appropriately for the readers to see them in these examples. Also, the criteria for the identification of S and crest synapses should be explained. 

3. Expression of dreadd and ChR2: The logic of injecting different viruses expressing cre and DIO-dreadd in the septum is unclear to me. What is the intended target of the virus expressing cre? A retroviral approach will allow for selectively targeting septal neurons that project to the mHb. Given that the majority of mHb neurons express Gfra1, what is the need for using a cre-based approach in the mHb as it excludes the use of WT group (cre-negative). More details are needed for how this study controls for the virus infection/expression of dreadd as these activity experiments involve challenging quantitative comparisons. Also, authors need to present CNO vs vehicle groups comparisons. Given the potential direct effect of CNO or its metabolites (Mahler and Aston-Jones, Neuropsychopharmacology, 43: 934-936, 2018), authors should consider the inclusion of dreadd-free CNO control experiments. 

4. The changes in calcium permeable AMPA receptors: Based on the data shown, the change in rectification index in this study is due to the modification of AMPA current at -60mV and not due to changes in AMPA current at depolarized conditions as all these groups appear to show significant inward rectification. Assuming that there is no noticeable inconsistency in reverse potential among groups/neurons and therefore no errors in the calculation, it is possible that both the increased amplitude at -60mV and an increased rectification index are due to high synaptic calcium permeable AMPA receptors in WT compared to HET and KO groups. A comparison of mEPSC amplitude might reveal whether there is an increase in AMPA receptor conductance while calculation of rectification index in cells (from these three groups) with similar EPSC amplitudes (-60mV) obtained after adjusting light intensity/duration will confirm these results. 

Biochemical studies show changes in AMPA receptor subunits in response to manipulation of Gfra1. However, the results are descriptive in nature and therefore, authors should avoid strong claims regarding the calcium permeability of AMPA receptors. Current results are far from establishing any mechanistic understanding of the role of GFRa1 in AMPA receptor regulation or septal-mHb-IPN function. 

5. Behavior: The foot-shock methods used in the current study and the interpretation of the results from these behavioral experiments are not entirely consistent with the current knowledge in the field. The lack of details in the method section and brief legends do not provide enough information to understand the behavioral procedures. The presented animal studies can only measure threat responses and not innate fear. Also, the descriptions in the manuscript are not clear about threat learning and memory. Panels in figures 6 and S7 are not clear about learning and memory. Authors also do not discuss the inconsistency in the data (Figures 6H and S7D). 

6. NMDA EPSC: NMDA EPSC is measured at 7 ms post-peak. Such an early decay of AMPA current is surprising (Adesnik et al., PNAS, 105 (14): 5597-602, 2008). Authors should show the data that AMPA currents are completely decayed at 7 ms. Alternatively, a comparison can be made at a later time point. 

7. Statistical analysis: The information about the number of mice (number of females and males)/slice/samples/cells in each group, the exact P and F values should be available to the reader. 

8. Discussion: A rigorous evaluation of the current results might show some important findings regarding the role of GRFa1 in the septal-mHb-IPN pathway. However, such an effort is lacking in the discussion. The discussion gives the impression that GFRa1 is important for fear memory regulation. Data show that an alteration of the septal-mHb-IPN pathway as a result of robust manipulation of GFRa1 affects the behavior. This does not mean a direct role of GFRa1 in these behaviors. Authors do not make an effort to discuss the basis of decreased anxiety and an increased threat response behavior. While it is acceptable to discuss the earlier studies showing similar behavior effects in response to specific manipulation of septal inputs, the current study using non-specific molecular manipulation of neurons is not comparable to those studies. 

Apart from the descriptive biochemical results, there are no clear results that mechanistically link GFRa1 to calcium permeability of AMPA receptors. Authors should also consider the fact that these data come from a possible heterogenous groups of neurons. Nevertheless, these results show that a diminished GFRa1 activity affects the AMPA receptor subunits. Therefore, authors should focus on the current results and avoid discussion about a direct role of GFRa1 in favoring the formation of calcium permeable AMPA receptors. 

9. References: Many statements throughout the manuscript lack references.

Minor comments

1. The study of PSD95 vs PSD93 needs to be clearly justified in the manuscript.

2. Fig S3b does not show DAPI.

3. Lateral IPN vs core IPN, details of core IPN should be included. 

Reviewer #2, Alberto Pascual: The paper by Fernández-Suárez et al. identifies the role of GFRa1 in synaptic stability and function. In general, the paper is very interesting, combining mouse models with viral injections that allow the authors to define the role of this protein in the septum-habenula-IPN circuit. The data are of quality and the experiments are well described. My opinion is very positive with several minor points:

1. Tamoxifen treatment may alter the behavior of mice. I only found in the M&M sections that all the groups were treated with TMX when describing the electron microscopy procedure. Please, specify in the mice section that all the groups (WT, Het, and KO mice) were treated with TMX. If that was not the case, at least behavioral experiments should be repeated with TMX treatment to all the experimental groups.

2. Quantification of electron microcopy images is not described in the M&M section, please include this information, including the number of synapsis analyze per mouse (also to be included in the figure legend). Regarding presynaptic vesicles/per vesicle pool, please define what is the meaning of "vesicle pool" (regarding the quantification). Was the quantification normalized to the area of the presynaptic boutons analyzed?

3. Regarding the analysis of the functional connectivity, line 258 should include the possibility of a role of GFRa1 during development (circuit establishment), as no differences were found between Het and KO mice (Fig. 4C). 

4. A trend is observed in SF5B and F regarding c-Fos expression, please add the p-value in those experiments (WT vs Het). This could suggest a developmental role of GFRa1 in the establishment of the circuit (in addition to the well-described phenotype already shown in other figures).

5. Please, also include the p-value in Fig. 4GI between WT and Het groups. To claim an adult role of GFRa1, the Het and the KO should be different, as the Het is the right control of the experiment (same genotype except in adult deletion of the Floxed allele). 

6. In Figure 5, I find the same problem, no differences are found between the WT and Het groups, but also no differences between Het and KO. In Fig. 5C-J, Het mice are directly excluded. Although that does not compromise the message of the article (Fig. 2E-H and Fig. 6 clearly demonstrate an adult role for GFRa1), the potential role of GFRa1 during development should be further discussed.

7. Our experience with antibodies against GDNF is not very positive. We have tested many of them using brain extracts from P0 WT and GDNF-/- mice and many of them recognize a band of 25 kDa present in both samples. I would kindly ask the authors to check the specificity of the GDNF antibody used and, if the band is not present in the P0 mutant extract, add the control to a supplementary figure to clarify that this is a perfect antibody. On the contrary, the figures with GDNF in western blot should be removed from the article and either substitute with ELISA or just show the q-RT-PCR data.

8. Line 120, this affirmation is controversial, see PMIDs: 18536709, 25710829, and 31930748.

9. Lines 139 and 152, please cite PMID: 27445711.

10. Please, homogenize through the text the way of presenting the p values (p < 0.00), as many different options are already stated (p<0.00; p <0.00; p< 0.00). Same thing with units (X mM instead of XmM).

11. In line 508, please change "H2O"

12. In line 609, please change "um" to "µm"

Reviewer #3: In this work, Fernandez-Suarez et al revealed the function of the GFRalpha1 in the medial habenula circuits using a multidisciplinary approach. Overall, I enjoyed the manuscript and all the data the authors were able to gathered. It is really a great work. I do have some concerns that I am sure the authors will be able to answer. 

About the AAVs injections in the MHb, the injection was bilateral but only in one anterior-posterior (AP) coordinate, right? If so, the deletion was observed along the full MHb? or only in one part? Did the AAVs reach other brain regions? is the expression of Gfra1 in the MHb constant along the AP MHb?? 

The authors checked the expression of any cholinergic marker in their experiments? If not? why? would you expect the same changes as the glutamatergic ones? 

In Figure 4B (cFos):in WT you have both dorsal and ventral Fos, in HET mostly dorsal, in KO mostly ventral (also maybe in the PVT). 

why?? is it juts my impression or is it constant? any double labelling planned? CHAT,vglut?

Page 15, line 347: why onlt 2 weeks? would you expect different results after 5 weeks for example? 

Page 15, line 348: nor sure if it is correct to say that you evaluate anxiety ? they are just the innate responses to potential danger. right? I mean, the behavior is not pathological. 

Do you have the distance measurement in the EPM? 

You mentioned that the KO mice show a trend towards increased distance travelled in the OF: it would be crucial to to make an analysis correlating the data obtained in the OF with the EPM (distance for example). To see if the mice "traveling" more are the same "traveling " more in the other tests. Because in some graphs it seems that you have two very distinct groups (e.g., latency in AA). <Is it because the expression off the Gfra1 is different in habenular circuits? Do you evaluate the expression the receptor after the behavioral paradigms? 

Did the authors evaluate pain perception? 

the authors didnt find differences in CFC, but there are differences in Passive avoidance? which also uses a CUE (am I right), which I think in this case is visual, right? so maybe there is a specifcity about the type of cue? does it make sense? 

Finally, did the authors try to evaluate any APPETITIVE conditioning (or response)? is it specific of AVERSIVENESS? 

Again, congratulations for the massive work.

---

## [Decision Letter · Decision Letter 2]

22 Sep 2021

Dear Dr Fernandez-Suarez,

Thank you for submitting your revised Research Article entitled "Adult medial habenula neurons require GDNF receptor GFRalpha1 for synaptic stability and function" for publication in PLOS Biology. I have now obtained advice from the original reviewers and have discussed their comments with the Academic Editor. 

Based on the reviews, we will probably accept this manuscript for publication, provided you satisfactorily address the remaining points raised by Reviewer 2. **IMPORTANT: Please also make sure to address the following data and other policy-related requests.

1) ETHICS REQUEST: Thank you for specifying the name of the ethics committee that reviewed and approved your animal care and use protocol. Please include the specific national or international regulations/guidelines to which your animal care and use protocol adhered. Please note that institutional or accreditation organization guidelines (such as AAALAC) do not meet this requirement.

2) BLURB: Please provide a blurb which (if accepted) will be included in our weekly and monthly Electronic Table of Contents, sent out to readers of PLOS Biology, and may be used to promote your article in social media. The blurb should be about 30-40 words long and is subject to editorial changes. It should, without exaggeration, entice people to read your manuscript. It should not be redundant with the title and should not contain acronyms or abbreviations. For examples, view our author guidelines: https://journals.plos.org/plosbiology/s/revising-your-manuscript.

3) DATA REQUEST: Please provide the underlying data for all relevant figures (including supplemental figures) as either a supplementary file or deposition in a publicly available repository. Note that we do not require all raw data. Rather, we ask that all individual quantitative observations that underlie the data summarized in the figures and results of your paper be made available.

Fig 1K,M; Fig 2B,D,F,H; Fig 3B,D; Fig 4 C,G-M; Fig 5A-F,H,J; Fig 6 A-B,D-E,G-H; Fig S1B-C,E; Fig S4D-H,K; Fig S5B,D,F,H; Fig S6B-D,G-H; Fig S7 B-C,E-F; Fig S8D-E; Fig S9A-H,J-K

**Please also ensure that figure legends in your manuscript include information on where the underlying data can be found, and ensure your supplemental data file/s has a legend.

**Please ensure that your Data Statement in the submission system accurately describes where your data can be found.

For more information on our data policy, see here: http://journals.plos.org/plosbiology/s/data-availability and here: http://dx.doi.org/10.1371/journal.pbio.1001797

We expect to receive your revised manuscript within two weeks. 

*Published Peer Review History*

*Early Version*

Sincerely,

Lucas Smith, Ph.D.,

Associate Editor,

lsmith@plos.org,

PLOS Biology

Reviewer remarks:

Reviewer #1: I have no further comments regarding this paper.

Reviewer #2, Alberto Pascual: The authors have addressed all my initial concerns. Two notes: 1) Western blot for GDNF should be removed in Fig1I, as the authors did not validate the antibody and the information is not relevant for the article. 2) In page 10-line 216, the sentence can be misleading, as no significant differences were found between HET and KO mice, so better remove "compared to Het or WT" and use "compared to WT".

Reviewer #3, Edgar Soria-Gómez: I have no further comments.

---

## [Editor Report · Decision Letter 3]

5 Oct 2021

Dear Dr Fernandez-Suarez,

On behalf of my colleagues and the Academic Editor, Eunjoon Kim, I am pleased to say that we can in principle offer to publish your Research Article "Adult medial habenula neurons require GDNF receptor GFRalpha1 for synaptic stability and function" in PLOS Biology, provided you address any remaining formatting and reporting issues. These will be detailed in an email that will follow this letter and that you will usually receive within 2-3 business days, during which time no action is required from you. Please note that we will not be able to formally accept your manuscript and schedule it for publication until you have made the required changes.

As you address the remaining formatting and reporting requests to come, we also ask that you address the following lingering editorial request: 

1) Thank you for providing the underlying for each of your figures on the Figshare database. *Please add a sentence to each figure legend (including supplemental) referencing this data. For example, you could add the sentence "the data underlying this figure can be found at https://figshare.com/projects/Raw_Data_Fernandez-Suarez_et_al_2021/123406"

PRESS

Sincerely, 

Lucas Smith, Ph.D. 

Senior Editor 

PLOS Biology

lsmith@plos.org